# Secondary Metabolites from Coral-Associated Fungi: Source, Chemistry and Bioactivities

**DOI:** 10.3390/jof8101043

**Published:** 2022-10-03

**Authors:** Ying Chen, Xiaoyan Pang, Yanchun He, Xiuping Lin, Xuefeng Zhou, Yonghong Liu, Bin Yang

**Affiliations:** 1CAS Key Laboratory of Tropical Marine Bio-resources and Ecology/Guangdong Key Laboratory of Marine Materia Medica, South China Sea Institute of Oceanology, Chinese Academy of Sciences, Guangzhou 510301, China; 2University of Chinese Academy of Sciences, 19 Yuquan Road, Beijing 100049, China

**Keywords:** coral-derived fungus, biological activities, natural products

## Abstract

Our study of the secondary metabolites of coral-associated fungi produced a valuable and extra-large chemical database. Many of them exhibit strong biological activity and can be used for promising drug lead compounds. Serving as an epitome of the most promising compounds, which take the ultra-new skeletons and/or remarkable bioactivities, this review presents an overview of new compounds and bioactive compounds isolated from coral-associated fungi, covering the literature from 2010 to 2021. Its scope included 423 metabolites, focusing on the bioactivity and structure diversity of these compounds. According to structure, these compounds can be roughly classified as terpenes, alkaloids, peptides, aromatics, lactones, steroids, and other compounds. Some of them described in this review possess a wide range of bioactivities, such as anticancer, antimicrobial, antifouling, and other activities. This review aims to provide some significant chemical and/or biological enlightenment for the study of marine natural products and marine drug development in the future.

## 1. Introduction

Marine organisms, representing approximately 75% of all living organisms, have proven to be a rich source of inspiration for drug discovery, with success rates for marine natural products up to 4 times higher than other naturally derived compounds [1]. Research into the pharmacological properties of marine natural products (MNP) has led to the discovery of many active agents considered worthy of clinical application; to date 14 marine NPs or their derivatives are registered drugs, and another 23 are currently in clinical trials [2]. The annual reviews of marine natural products were reported by the New Zealand group in the Natural Product Reports. These reviews show that marine fungi are currently the most studied marine microorganism phyla, and over the last five years an extraordinary transformation in MNP research continued with a very significant increase in the number of new compounds reported from marine fungi. For example, in 2018, new MNPs reported from marine fungi increased by 38% relative to 2017 [3]. With up to 90% of marine species undescribed, marine fungi can inspire new discoveries and offer many novel solutions to life’s problems in the future.

Coral reefs are among the most fragile, biologically diverse and economically important ecosystems on Earth, providing ecosystem services that are vital to human societies and industries through fisheries, coastal protection, new biochemical compounds, and tourism [4,5]. Coral reefs are regarded as one of the most important shelters of microorganisms [6]. Fungi are abundant in the coral reefs and recent studies demonstrate diverse communities associated with coral. Recent advances suggest that fungi associated with marine invertebrates may play an active role in the formation of biofilms and constitute the main chemical defense mechanism of the host. It is widely accepted that small molecule natural products evolved to carry out a particular ecological function and that these finely tuned compounds can sometimes be appropriated for the treatment of disease in humans [7].

Since the 1960s, plenty of bioactive metabolites have been isolated from coral. However, the supply has become a serious obstacle to the ultimate development of these bioactive substances. This has given rise to investigations on the metabolites produced by coral-symbiotic microorganisms. Two reviews on the biological and chemical diversity of coral-associated microorganisms were published by Shao and his coworkers [8,9]. In 2018, Keller-Costa et al. studied how bioactive secondary metabolites form octocoral-associated bacteria and fungi [10]. In 2021, Seelan et al. compiled a review to summarize metabolites produced by marine fungi isolated from the soft coral genus *Sarcophyton* from 2010 to 2020 [11]. Similar to these prior reviews, they focused on the discovery of new compounds from natural sources and the associated biological properties of these metabolites. In this review, we provide a thorough overview of new compounds and bioactive metabolites gathered from coral-derived fungi from 2010 to 2021, which present 423 compounds of chemical structures and bioactivities.

## 2. Terpenes

Terpenes have been widely applied in the pharmaceutical, nutraceutical, synthetic chemistry, flavor fragrance, and possibly biofuel industries, and are essential constituents of natural products. Moreover, terpenes are a prime group of essential oils possessing a broad spectrum of antibacterial, antifungal and even antiviral activity [12,13]. Marine fungi are a significant source of terpenes, so it is necessary to carry out further investigation.

### 2.1. Sesquiterpenes

Previous research revealed that approximately 500 new sesquiterpenes, including about 20 new skeletons, were characterized from fungi during a five-year period (2015~2020) [14]. Soft coral-associated fungi are reported to be a rich source of sesquiterpenes. Li et al. have obtained a series of sesquiterpenoids from soft-coral associated fungi. Eleven linear triquinane sesquiterpenoids including chondrosterins A–E (**1**–**5**) [15], chondrosterin F (**6**) and incarnal (**7**) [16], hirsutanol A (**8**), [17] hirsutanol E (**9**), chondrosterin N (**10**) and chondrosterin O (**11**) [18] were isolated from the fungus *Chondrostereum* sp., separated from the soft coral *Sarcophyton tortuosum*, from the South China Sea. In addition, a new aromadendrane sesquiterpenoid, pseuboydone F (**12**), was extracted from the fungus *Pseudallescheria boydii* F44-1, derived from the soft coral *Sarcophyton* sp. [19]. Two new aromadendrane-type sesquiterpene diastereomers, pseuboydones A (**13**) and B (**14**), were also discovered from the marine-derived fungus *Pseudallescheria boydii* F19-1, collected from the soft coral *Lobophytum crissum* [20]. Linear triquinane sesquiterpenoids, possessing a basic skeleton 1H-cyclopenta[α]pentalene, have been isolated from different organisms including fungi, sponges and soft coral. In 2018, Li et al. published an overview covering 118 linear triquinane sesquiterpenoids [21]. From a biomedical perspective, chondrosterin A (**1**) with the typical α-methylene ketone group revealed significant cytotoxic activities against cancer lines A549, CNE2, and LoVo with IC_50_ values ranging from 2.45 to 5.47 μM, and chondrosterin B (**2**) exhibited antimalarial activity with an IC_50_ value of 3.10 μg/mL [22]; incarnal (**7**) had potent cytotoxic activity against cancer cell lines (CNE1, CNE2, SUNE, LoVo, KB, Be17402, MCF-7) with IC_50_ values ranging from 2.16 to 28.55 μg/mL; hirsutanol A (**8**) showed pronounced cytotoxic activities against various cancer cell lines (SW620, SW480, LoVo; Hep3B, HepG2, Bel-7402; A549; CNE1, CNE2, SUNE1; MCF7, MDA-MB-231, MDA-MB-435, MDA-MB-453 and HeLa) with IC_50_ values ranging from 0.58 to 8.27 μg/mL. The chemical exploration of the gorgonian-derived fungus *Aspergillus* sp. was carried out and afforded three known sesquiterpenoids, (*R*)-(-)-hydroxysydonic acid (**15**), (*S*)-(-)-5-(hydroxymethyl)-2-(2’,6’,6’-trimethyltetrahydro-2*H*-pyran-2-yl) phenol (**16**), and (*S*)-(+)-11-dehydrosydonic acid (**17**), which revealed moderate antibacterial activity against *Staphylococcus*
*aureus, Bacillus cereus, Kocuria rhizophila, Pseudomonas putida, Pseudomonas aeruginosa, Salmonella enterica,* and *Nocardia. brasiliensis* [23] (Figure 1).

### 2.2. Diterpenoids

The well-known compound lovastatin (**18**) was extracted from a coral-derived fungus *Aspergillus terreus*. In vitro anti-inflammatory experiments showed that **18** has anti-inflammatory activity against NO production and was shown to have a significant inhibitory effect with an IC_50_ value of 17.45 μM [24]. Meanwhile, compound **18** was an inhibitor of 3-hydroxy-3-methyl-glutaryl-coenzyme as a lipid-lowering drug [25]. Furthermore, compound **18** also exhibited bioactivities of anti-cancer, prevention and treatment of neurological disorders, and antibacterial effects [26]. Two new harziane diterpene lactones, harziane lactones A and B (**19** and **20**) and five new harziane diterpenes, harzianones A–D (**21**–**24**) and harziane (**25**), were identified from the soft coral-derived fungus *Trichoderma harzianum* XS-20090075. Compounds **19**–**23** and compound **25** exhibited obvious phytotoxicity against the seedling growth of amaranth and lettuce with a concentration of 200 ppm. Moreover, at the concentration of 200 μg/mL, compounds **19**, **21**, **22**, and **23** completely inhibited seed germination against amaranth. Compared with the positive control glyphosate, these compounds still showed phytotoxicity at a lower concentration of 50 μg/mL [27] (Figure 2).

### 2.3. Triterpenes

A new nordammarane triterpenoid (**26**) was characterized from the marine strain *Aspergillus fumigatus* KMM 4631 collected from the soft coral *Sinularia* sp. [28] (Figure 3).

### 2.4. Meroterpenoids

Fungi are the remarkable producers of meroterpenoids, which have exhibited diversified and unique structures with a wide range of bioactivities [29]. Marine-derived fungus *Aspergillus* sp. was obtained from the gorgonian *Dichotella gemmacea* collected from the South China Sea, which produced three new phenolic bisabolane-type sesquiterpenoids: (+)-methyl sydowate (**27**), 7-deoxy-7, 14-didehydrosydonic acid (**28**), and 7-deoxy-7,8-didehydrosydonic acid (**29**) [30]. Compound **27** showed bioactivity against *S. aureus* and methicillin-resistant *Staphylococcus*
*aureus* (MRSA) with the same inhibition zones of 11 mm in diameter at a concentration of 100 μg/mL, while the positive control had inhibition zones of 37 and 21 mm in diameter, respectively. Moreover, compound **28** showed remarkable inhibitory activity on *Gaeumannomyces graminis* (MIC = 0.5 μg/mL) [31]. The chemical investigation of *Aspergillus versicolor* (ZJ-2008015) led to the isolation of three bisabolane sesquiterpene compounds named (+)-sydonic acid (**30**), expansol G (**31**), and (+)-sydowic acid (**32**). An in vitro antibacterial experiment showed that all compounds exhibited strong antibacterial activity with the MIC values of 5.3, 6.4, and 5.4 μg/mL, respectively, for *Staphylococcus albus* and 2.6, 6.4, and 5.4 μg/mL, respectively, for *S. aureus* [32]. Moreover, compound **32** also revealed cytotoxicity against murine leukemia P-388 cells (IC_50_ = 2.56 μg/mL) [33]. Two new phenolic bisabolane-type sesquiterpenes, namely 11,12-dihydroxysydonic acid (**33**) and 1-hydroxyboivinianic acid (**34**), were extracted from the marine-derived fungus *Scopulariopsis* sp. collected from the Red Sea hard coral *Stylophora* sp. [34]. Meanwhile, compound **34** exhibited weak activity to cancer cell lines A549, Caski, HepG 2, and MCF-7 with IC_50_ values of 90.6, 78.2, 75.8, and 80.4 μg/mL, respectively. Furthermore, **34** also showed antimicrobial activity against *Erwinia carotovora* sub sp. carotovora (MIC = 68.9 μg/mL) [35]. Craterellin D (**35**), a new merosesquiterpenoid, together with its known analog craterellin A (**36**), was isolated from a soft coral-derived *Lophiostoma* sp. fungus. Compound **35** was the ramification of the compound **36**, whose configuration was confirmed by modified Mosher’s method and single-crystal X-ray diffraction. A primary bioassay indicated that **36** was shown to have antibacterial activity against *B. cereus* (MIC = 3.12 μM) [36] (Figure 4).

## 3. Alkaloids

Alkaloids have great development potential as drug scaffolds and scaffold substructures in modern antibacterial chemotherapy. There will be opportunities to accelerate the process of discovering more active alkaloids in the future [37].

### 3.1. Diketopiperazines

Wang et al. isolated three novel diketopiperazine alkaloids, chrysopiperazines A–C (**37–39**), from gorgonian-derived fungus *Penicillium chrysogenum* [38]. In addition, Wang’s group also discovered a strain of *Nigrospora oryzae* (ZJ-2008005) derived from soft coral *Dendronephthya* sp., which was found to produce seven isoechinulin-type alkaloids named neoechinulin A (**40**), preechinulin (**41**), isoechinulin A (**42**), tardioxopiperazine A (**43**), variecolorin L (**44**), dihydroxyisoechinulin A (**45**), and *L*-alanyl-*L*-tryptophan anhydride (**46**). A primary bioassay of antifouling activity against the larval settlement of barnacle *Balanus amphitrite* showed that compound **42** revealed significant antifouling activity with an IC_50_ value of 5.92 µg/mL, while **40** and **45** presented weak activity with IC_50_ values of 30 and 50 µg/mL, respectively [39]. Moreover, compound **40** also displayed strong cytotoxic activity against human cervical carcinoma HeLa cells and exerted anti-inflammatory, antiviral, and neuroprotective activity [40,41]. Furthermore, compound **42** revealed inhibitory activities against AChE (IC_50_ = 74.38 µM), and compounds **40** and **42** demonstrated the DPPH-scavenging effect at IC_50_ 44.30 and 103 µM, respectively, while **41**, **43**, and **44** were inactive at IC_50_ > 500 µM [42,43]. In short, **40** was a bioactive compound. Compound **43** exhibited moderate immunosuppressive activity of Con A-induced and LPS-induced with IC_50_ values of 4.5 and 0.7µM, respectively [44] and compounds **43** and **44** showed very weak cytotoxic effects to NCI-H1975/GR cell lines with the inhibition ratio of 50.94% and 56.83%, respectively, at a concentration of 50 μM [45]. A recent study showed that compound **46** revealed inhibitory activity against TMEM16A with 65% inhibition with a concentration of 5 µg/mL) [46]. Another study found seven new isoechinulin-type alkaloids named 16α-hydroxy-17*β*-methoxy-deoxydihydroisoaustamide (**47**), 16*β*-hydroxy-17*α*-methoxy-deoxydihydroisoaustamide (**48**), 16*α*-hydroxy-17*α*-methoxy-deoxydihydroisoaustamide (**49**), 16,17-dihydroxy-deoxydihydroisoaustamide (**50**), 16*β*,17*α*-dihydroxy-deoxydihydroisoaustamide (**51**), 16*α*,17*α*-dihydroxy-deoxydihydroisoaustamide (**52**), and 3*β*-hydroxy-deoxyisoaustamide (**53**) isolated from the coral-derived fungus *Penicillium dimorphosporum* KMM 4689. On these compounds, neuroprotective activity experiments were conducted. The results indicated that compounds **50** and **52** with a concentration of 1 µM could improve the viability of PQ (paraquat)-treated (PQ = 500 µM) Neuro-2a cells by 38.6% and 30.3%, respectively, while compound **51** increased the cell survival rate by 36.5% and 39.4% at concentrations of 1 μM and 10 µM, respectively [47]. Three new indole diketopiperazine alkaloids named 11-methylneoechinulin E (**54**), variecolorin M (**55**), and (+)-variecolorin G (**56**), together with a known compound, (-)-variecolorin G (**57**), were extracted from a soft coral-associated epiphytic fungus *Aspergillus* sp. EGF 15-0-3. Compound **57** showed very weak cytotoxic effects to NCI-H1975/GR cell lines with the inhibition ratio of 40.65% at a concentration of 50 μM [45]. Seven notoamide-type alkaloids were obtained from a culture broth of coral-associated fungus *Aspergillus ochraceus* LZDX-32-15, including four new congeners, namely notoamides W-Z (**58–61**) and three reported alkaloids, namely notoamide G (**62**), avrainvillamide (**63**), and stephacidin B (**64**), which displayed inhibition against a panel of hepatocellular carcinoma (HCC) cell lines with IC_50_ values ranging from 0.42 to 3.39 μM. Compared with the positive control of paclitaxel with a concentration of 10 µM, these compounds also revealed notable growth inhibition against tumor cell lines (human HCC cell line SMMC-772, human colorectal carcinoma cell line HCT-8, human breast cancer cell line MCF-7, and human umbilical vein endothelial cell HUVEC) with an inhibition ratio of more than 95% [48]. In addition, compound **63** displayed considerable antiproliferative activity and reversible inhibition against human GR activity in LNCaP cell lysate (IC_50_ = 125 µM). But dramatically, compounds **63** and **64** were shown to be a new class of toxins with potential threat to human health in buildings [49,50]. A novel compound pseudellone D (**65**) was obtained from the marine-derived fungus *Pseudallescheria ellipsoidea* F42-3 collected from the soft coral *Lobophytum crissum* [51]. Two novel metabolites diketopiperazines pseuboydones C, D (**66, 67**) and a known compound cyclo-(Phe-Phe) (**68**) were produced by the marine-derived fungus associated with the soft coral *Lobophytum crissum.* The compound **68** showed strong activity against cell line Sf9 from the fall armyworm *Spodoptera frugiperda* (IC_50_ = 0.8 µM) [20]. A described compound named cyclo-(Pro-Val) (**69**) was characterized from the fungus *Simplicillium* sp. SCSIO 41209 gathered from soft coral and exhibited inhibition to MptpB (IC_50_ = 25.9 μM) [52] (Figure 5).

### 3.2. Quinazolinone Alkaloids

Gorgonian-derived fungus *Aspergillus versicolor* was considered to be a great potential producer of quinazolinone alkaloids. Cheng et al. conducted research on the gorgonian-derived fungus *Aspergillus versicolor* LZD-14-1 resulting in the identification of four described metabolites and six new polycyclic alkaloids, namely versiquinazolines A, B, G, and K (**70**–**73**) [53] and versiquinazolines L–Q (**74–79**) [54]. Metabolites **76–79** displayed TrxR inhibitory activity with IC_50_ values of 20, 12, 13, and 13 μM, respectively [53]. Bioassay results showed that **74–79** exhibited weak activity against cell line A549 at IC_50_ > 10 μM. Moreover, versiquinazolines P (**78**) and Q (**79**) also demonstrated significant inhibition against TrxR with IC_50_ values of 13.6 μM and 12.2 μM, respectively, while curcumin (positive control group) showed an IC_50_ value of 25 μM [54]. In addition, three new quinazolinone alkaloids, cottoquinazolines B-D (**80–82**), were produced from *Aspergillus versicolor* LCJ-5-4. The bioassay test indicated that only **82** revealed middling antifungal activity against *Candida albicans* (MIC = 22.6 μM) [55]. A previously unreported alkaloid, namely fumiquinazoline K (**83**) was discovered from a marine strain of *Aspergillus fumigatus* KMM 4631 [28]. Studies on gorgonian-derived fungus *Scopulariopis* sp. led to the isolation of fumiquinazoline L (**84**) which showed weak antibacterial activity against pathogenic bacteria *B. subtilis*, *S*. *albus*, and *Vibrio parahemolyticus* (MIC = 50 μM) [56]. Two new quinazolinone analogues, namely 4-(7,8-Dihydroxy-4-oxoquinazolin-3(4H)-yl), butanoic acid (**85**) and 4-(8-Hydroxy-4-oxoquinazolin-3(4H)-yl) butanoic acid (**86**), were isolated from the fungus *Xylaria* sp. FM1005, which was gathered from a leather coral *Sinularia densa* found in the area of the Big Island, Hawaii [57]. Further investigation of *Neosartorya laciniosa* (KUFC 7896) led to the isolation of a novel tryptoquivaline derivative, tryptoquivaline T (**87**) [58]. Two new alkaloid racemates, (±)-17- hydroxybrevianamide N (**88**&**89**) and (±)-N1-methyl-17-hydroxybrevia-namide (**90**&**91**), possessing a peculiar *O*-hydroxyphenylalanine residue and an imide subunit were obtained from a soft coral-derived *Aspergillus* sp. [59]. Another two new alkaloids namely pseudellones A (**92**) and B (**93**) were produced by the marine-derived fungus *Pseudallescheria ellipsoidea* F42−3, which was collected from a soft coral *Lobophytum crissum* [60] (Figure 6).

### 3.3. Cytochalasins

Previous chemical exploration on the *Chaetomium globosum* C2F17 led to the separation of 6-*O*-methyl-chaetoglobosin Q (**94**), chaetoglobosin E (**95**), and chaetoglobosin Fex (**96**). Furthermore, in an in vitro anticancer experiment, **95** exhibited strong inhibitory activity against cell lines HCT-116, K562, A549, Huh7, H1975, MCF-7, U937, BGC823, HL60, HeLa, and MOLT-4, with IC_50_ values of 13.5, 8.9, 5.9, 1.4, 9.2, 2.1, 1.4, 8.2, 2.5, 2.8, and 1.4 µM, respectively. Compound **96** revealed selective cytotoxic activity against cell lines Huh7, MCF-7, U937, and MOLT-4, with IC_50_ values of 3.0, 7.5, 4.9, and 2.9 µM, respectively, and also exhibited anti-inflammatory activity [61,62,63]. *Aspergillus elegans* ZJ-2008010, isolated from a soft coral *Sarcophyton* sp. in the South China Sea, produced seven cytochalasins, namely aspochalasin A1 (**97**), cytochalasin Z24 (**98**), aspochalasin I (**99**) aspochalasin J (**100**), aspochalasin D (**101**), aspochalasin H (**102**), and aspergillin PZ (**103**). According to the experimental data of biological activity assay, **99**–**102** showed notable antifouling activity to the larval settlement of the barnacle *Balanus amphitrite* with EC_50_ values of 34, 14, 6.2, and 37 μM, respectively. Moreover, **101** also presented antibacterial activity against *S. albus*, *S*. *aureus*, *Escherichia coli,* and *B. cereus* (MIC = 10 μM), while compound **99** was found to have weak activity against *Staphylococcus epidermidis* and *S. aureus* with MIC values of 20 and 10 μM, respectively; and compound **103** as well showed very weak activity against *S. epidermidis* (MIC = 20 μM). Furthermore, compound **99** possessed potential inhibition to melanogenesis in Mel-Ab cells (IC_50_ = 22.4 μM) [64]. Moreover, compound **103** displayed DPPH free radical-scavenging effects increasing with concentrations and anticancer activity against the A2780, LNCaP, and PC3 cell lines [65,66]. Another three cytochalasins, chaetoglobosins A and B (**104** and **105**), and cytoglobosin C (**106**) were obtained from the fungus *Chaetomium globosum* RA07-3 [67]. Compounds **104** and **105** exerted outstanding inhibition against *Tetragenococcus halophilus* with MIC values of 0.7 and 0.4 μM, respectively. Compound **106** also showed antibacterial activity against *T. halophilus* (MIC = 0.7 μM). Meanwhile, this group also obtained aspochalasin K (**107**) and aspochalasin E (**108**) from *Aspergillus* sp. XS-2009-0B15. In bioassays, **107** revealed antibacterial activity, while **108** showed anticancer activity against B16-F10 and HCT-116 with IC_50_ values of 18.5 and 6.3 μg/mL, respectively [68,69] (Figure 7).

### 3.4. Other Alkaloids

The gorgonian coral-derived fungus *Scopulariopsis* sp. was regarded as a producer of five dihydroquinolin-2-one-containing alkaloids including three monoterpene alkaloids [70] and a new alkaloid. They were 4-phenyl-3, 4-dihydroquinolin-2(1H)-one named aniduquinolone A (**109**), aflaquinolone A (**110**), aflaquinolone D (**111**), and two 4-phenyl-3,4-dihydroquinolin-2(1H)-one named 6-deoxyaflaquinolone E (**112**) and aflaquinolone F (**113**), and a new metabolite named scopuquinolone B (**114**) [71]. All compounds (**109–114**) exhibited notable antifouling activity against the settlement of *B. amphitrite* cyprids with EC_50_ values of 17.5 pM, and MIC values of 28 nM, 2.8 nM, 1.04 μM, 0.86 μM, and 0.103 μM, respectively. Scopuquinolone B (**114**) possessed a high therapeutic ratio (LC_50_/EC_50_) of 222, which is stronger than the positive control Sea Nine 211 (EC_50_ = 4.36 μM, LC_50_/ EC_50_ = 20). Besides, no cytotoxicity was found in **109** or **110**; thus, they could be considered as new nontoxic anti-larval settlement compounds, especially **109** as a potential antifouling lead compound, which possessed safety and a high therapeutic ratio (LC_50_/EC_50_ = 1200) in nature. Another study revealed that the epimers of compound **109** exhibited considerable antibacterial activity against *S. aureus* (ATCC700699) [72]. Compound **110** showed lethality against brine shrimp (*Artemia salina*, LD_50_ = 5.5 μM) [73]. Furthermore, **112** revealed broad antibacterial spectrum of *S. aureus, B. cereus, Vibrio parahaemolyticus, N. brasiliensis,* and *P. putida*, with MIC values of 0.78, 1.56, 6.25, 0.78, and 1.56 μM, respectively. Five new compounds were found in the associated-coral fungus *Xylaria* sp. FM1005, namely sinuxylamide A (**115**), sinuxylamide B (**116**), sinuxylamide C (**117**), sinuxylamide D (**118**), and assinuxylamide E (**119**). Compounds **115** and **116** also showed significant inhibition against the binding of fibrinogen to purified integrin IIIb/IIa in a dose-dependent manner with IC_50_ values of 0.89 and 0.61 μM, respectively [57]. A new compound pseuboydone E (**120**) and three described compounds, namely speradine C (**121**), 24, 25-dehydro-10,11-dihydro-20-hydro-xyaflavinin (**122**), and aflavinine (**123**), were found in a fungus associated to *Lobophytum crissum*. All known compounds revealed notable inhibition activity against the Sf9 cells with IC_50_ values of 0.9, 0.5, and 0.4 μM, respectively. Compounds **122** and **123** also showed similar significant cytotoxicity to the positive control, rotenone [20]. Ten alkaloids were produced by coral-derived fungi *Aspergillus terreus* including eight new alkaloids: aspergillspins A-E (**124**–**128**) [74], luteoride E (**129**) [24], 22-*O*-(*N*-Me-*L*-valyl) aflaquinolone B (**130**), and 22-*O*-(*N*-Me-*L*-valyl)-21-*epi*-aflaquinolone B (**131**) [75], and two known metabolites asperteramide A (**1****32**) and methyl 3,4,5-trimethoxy-2-(2-(nicotinamido) benzamido) benzoate (**133**). In addition, compound **131** revealed excellent anti-respiratory syncytial virus activity (IC_50_ = 42 nM) [75]. Moreover, compound **132** displayed evident antimicrobial activity against *E. coli, Acinetobacter baumannii, P. aeruginosa, Klebsiella pneumonia*, MRSA*,*
*Enterococcus*
*faecalis*, and *C. albicans* with MIC values of 8, 8, 16, 64, 64, 8, 2 μg/mL, respectively [76]. In addition, luteoride E (**129**) and methyl 3,4,5-trimethoxy-2-(2-(nicotinamido) benzamido) benzoate (**133**) exerted anti-inflammatory activity against NO production with IC_50_ values of 24.64 and 5.48 μM, respectively. Neoaspergillic acid (**134**), also derived from fungus *Aspergillus* sp. SCSGAF0093, exhibited bio-toxicity against the brine shrimp (*Artemia salina*) with an IC_50_ value of 90.08 µM and also demonstrated anticancer activity against SPC-A-1, BEL-7402, SGC-7901, and K562 with IC_50_ values of 22.2, 24.9, 8.2, and 8.0 µM, respectively. Moreover, compound **134** possessed antibacterial activity (*S. aureus, S. epidermidis, B. subtilis, Bacillus dysenteriae, Bacillus proteus, E. coli*) with a MIC of 1.0, 0.5, 1.9, 7.8, 7.8, and 15.6 μg/mL, respectively [77,78]. A new natural product, ethyl 2-bromo-4-chloroquinoline-3- carboxylate (**135**), was isolated from soft coral-associated fungus *Trichoderma harzianum* (XS-20090075), which was the first halogenate quinoline metabolite from the genus *Trichoderma* [79]. An undescribed alkaloid, namely scopulamide (**136**) was derived from the marine-derived fungus *Scopulariopsis* sp., which was found in the Red Sea hard coral *Stylophora* sp. [34]. Pyrophen (**137**) was identified from the *Alternaria alternata* strain D2006 associated with soft coral and was shown to be active against *C*. *albicans* with the inhibition zone 28 mm [80]. One undescribed phenylspirodrimane derivative, named arthproliferin E (**138**), was isolated from the soft coral-associated fungus *Stachybotrys chartarum* SCSIO41201 [81]. (3*R*,6*R*)-bassiatin (**139**) was isolated from the soft coral-derived fungus *Dichotomomyces* sp. L-8 and displayed significant cytotoxic activities against the human breast cancer cell line MDA-MB-435 and human lung cancer cell line Calu3 with IC_50_ values of 7.34 and 14.54 µM, respectively [82] (Figure 8).

## 4. Peptides and Depsipeptides

### 4.1. Cyclopeptides

A study reported that cyclopeptides displayed high effects against various cancer cells, and some of them were even applied in clinical experiments [83]. Previous chemical exploration of *Aspergillus versicolor* led to the separation of 11 new cycloheptapeptides, namely, asperheptatides A−D (**140−143**) [84], versicoloritides A-C (**144–146**) [85], versicotides D–F (**147–149**) [86], and aspersymmetide A (**150**) [87]. Versicotides D–F (**147–149**) showed anti-atherosclerosis activity [86]. Aspersymmetide A (**150**) exerted weak anticancer activity against NCI-H292 and A431 with an inhibition ratio of 53.8% and 63.62% (10 μM), respectively [87]. *Penicillium chrysogenum* (CHNSCLM-0003), isolated from gorgonian coral, produced seven new cyclohexadepsipeptides, chrysogeamides A–G (**151–157**). Furthermore, compounds **151** and **152** were shown to have angiogenic activity toward zebrafish embryo (IC_50_ = 1.0 μg/mL) with a non-cytotoxic concentration of 100 μg/mL [88]. A new compound sinulariapeptide A (**158**) and three described compounds, simplicilliumtides A (**159**), B (**160**) and J (**161**) were derived from the soft coral-associated fungus *Simplicillium* sp. SCSIO 41209. In the bioassays, compounds **158**–**161** showed inhibition to *Colletotrichum asianum* with the MIC values of 4.9, 19.5, 4.9, and 9.8 μg/mL, respectively, while a positive control (actidione) with a MIC value of 10 μg/mL. Moreover, compound **160** was found to have notable inhibition to *Pyricularia oryzae*, while **159** and **161** exhibited weak inhibitory activities with the MIC values of 9.8, 19.5, and 78.1 μg/mL, respectively [52]. Scopularide A (**162**) was obtained from the fungus *Scopulariopsis* sp. Collected from the Red Sea hard coral *Stylophora* sp., which revealed remarkable cytotoxicity to the growth of the murine lymphoma cell line (L5178Y) (IC_50_ = 1.2 µM) [34]. Two new metabolites, asperpeptide A (**163**) [89] and aspergillipeptide D (**164**) [90], were isolated from the coral-derived fungus *Aspergillus* sp. Asperpeptide A (**163**) displayed antibacterial activity (*B. cereus* and *S. epidermidis*) with the same MIC value of 12.5 μM [89], while aspergillipeptide D (**164**) exhibited considerable antiviral activity against HSV-1 (IC_50_ = 9.5 μM) [90] (Figure 9).

### 4.2. Linear Peptides

Four new compounds named sinulariapeptides B−E (**165–168**), together with a known metabolite hirsutellic acid A (**169**), were isolated from a culture broth of the soft coral-derived fungus *Simplicillium* sp. SCSIO 41209. Hirsutellic acid A (**169**) showed inhibition to MptpB (IC_50_ = 35.0 µM) [52]. (+) - and (−) - pestaloxazine A (**170**, **171**), a pair of new enantiomeric alkaloid dimmers, was derived from a *Pestalotiopsis* sp., which have a new symmetric spiro-(oxazinane-piperazinedione) skeleton [91]. Compound **170** exhibited significant antiviral activity to EV71 (IC_50_ = 14.2 μM) stronger than the positive control ribavirin (IC_50_ = 256.1 μM). In addition, a new compound named pseudellone C (**172**) was found in the fungus *Pseudallescheria ellipsoidea* F42−3, and **172** was found for the first time in previous literature possessing a unique skeleton [60]. The genus *Aspergillus* was widely investigated and yielded a variety of metabolites. Thirteen compounds were isolated from coral-derived *Aspergillus* sp.: namely 4’-OMe-asperphenamate (**173**), asperphenamate (**174**), aspergilliamide (**175**), ochratoxin A n-butyl ester (**176**), flavacol (**177**), ochratoxin A (**178**) and ochratoxin A methyl ester (**179**), penilumamides B-D (**180**–**182**), and aspergillipeptides E–G (**183**–**185**). 4’-OMe-asperphenamate (**173**), a new phenylalanine derivative, along with a known phenylalanine derivative asperphenamate (**174**), was isolated from *Aspergillus elegans* ZJ-2008010, which was collected from a soft coral *Sarcophyton* sp. in the South China Sea. Both of them exhibited selective antibacterial activity against *S. epidermidis*, with a MIC value of 10 μM for each [65]. A novel and four known metabolites, aspergilliamide (**175**), ochratoxin A n-butyl ester (**176**), flavacol (**177**), ochratoxin A (**188**), and ochratoxin A methyl ester (**189**) were obtained from the strain *Aspergillus* sp. SCSGAF0093. Compounds **175**–**179** showed bio-toxicity against the brine shrimp (*Artemia salina*) with LC_50_ values of 71.09, 4.14, 205.67, 13.74, and 2.59 µM, respectively. Flavacol (**177**) also displayed inhibition against NADH oxidase (IC_50_ = 13.0 μM). Meanwhile, ochratoxin A (**178**) was reported as a potential nephrotoxin and latent human carcinogen [77,92,93]. *Aspergillus* sp. XS-20090B15 was the source of three new peptides, penilumamides B-D (**180**–**182**) [89]. Moreover, *Aspergillus* sp. SCSIO 41501 also yielded three novel linear peptides, aspergillipeptides E–G (**183**–**185**), and compound **183** displayed antiviral activity against HSV-1 (IC_50_ = 19.8 μM) [90] (Figure 10).

## 5. Aromatics

### 5.1. Polyketides

The following study is of great significance for some antifouling active compounds. Fourteen new polyketides named libertalides A−N (**186**–**199**), along with two known analogues aspermytin A and its acetate (**200**, **201**) were produced by the coral-associated fungus *Libertasomyces* sp. In the bioassays, **187**, **193**, and **200** significantly induced the proliferation of CD3^+^ T cells with a concentration of 3 μM. Meanwhile, **190**, **194**, **197**, and **201** obviously increased the CD4^+^/CD8^+^ ratio with 3 μM. It was the first report in the immunoregulatory activity of these metabolites [94]. In addition, aspermytin A (**200**) was found with activity against neurite outgrowth in rat pheochromocytoma (PC-12) cells (50 μM) [95]. Two new C12 polyketides (**202**, **203**) with four known derivatives (**204**–**207**) were produced from a soft coral-derived fungus *Cladosporium* sp. TZP-29. They were named cladospolides E and F (**202** and **203**), secopatulolides A and C (**204** and **205**), 11-hydroxy-γ-dodecalactone (**206**), and iso-cladospolide B (**207**). Moreover, **202**, **204**–**206** exerted notable lipid-lowering activity in HepG2 hepatocytes with IC_50_ values of 12.1, 8.4, 13.1, and 7.1 μM, respectively [96]. Four new polyketides, including an unusual naphthoquinone derivative sclerketide A (**208**), two azaphilone analogous sclerketides B (**209**) and C (**210**), and an α-pyrone derivative sclerketide D (**211**), together with two known compounds, namely isochromophilone IX (**212**) and sequoiatone B (**213**), were isolated from the gorgonian-derived fungus *Penicillium sclerotiorum* CHNSCLM-0013. Compounds **209**–**213** exhibited a potential inhibitory effect against the production of NO with the IC_50_ values of 3.4, 2.7, 5.5, 17.6, and 5.2 μM, respectively. As the most active compound, **210** also showed moderate cytotoxicity (IC_50_ = 14.8 μM) [97]. Some other new antibacterial chloro-containing polyketides were identified from alga-derived fungus [98]. Furthermore, isochromophilone IX (**212**) presented outstanding antibacterial activity against MRSA (MIC = 50 μg/mL) [99]. The marine strain *Penicillium glabrum* glmu003 yielded two novel azaphilone compounds, daldinins G (**214**) and H (**215**), and a known compound austalide V (**216**). Moreover, austalide V (**216**) exhibited weak inhibitory activity against pancreatic lipase (IC_50_ = 23.9 μg/mL) [100]. A new asteltoxin B (**217**) was obtained from the gorgonian-associated fungus *Aspergillu*s sp. SCSGAF 0076 and revealed inhibition to human acetylcholinesterase (IC_50_ = 14.9 μM) [101,102]. Wang et al. found eight polyketides from gorgonian coral-derived fungus, namely aspergilone A (**218**), pinophilins D−F (**219**–**221**) [103], (+)-sclerotiorin (**222**), microketides A and B (**223** and **224**), and *epi*-pinophilin B (**225**). A novel compound, aspergilone A (**218**), was isolated from the fungus *Aspergillus* sp. and displayed antifouling activity against *Balanus amphitrite* (EC_50_ = 25 μg/mL) [104]. *Penicillium pinophilum* XS-20090E18 produced three new metabolites, pinophilins D−F (**219**–**221**). A bioactive azaphilone derivative, (+)-sclerotiorin (**222**) was also discovered from the *Penicillium sclerotiorum* fungus and possessed a wide range of bioactivities, which inhibited the larval settlement of barnacle *Balanus amphitrite* (EC_50_ = 5.6 μg/mL) and showed evident inhibition to *B. subtilis*, *B. cereus*, and *Sarcina lutea* with MIC values of 0.16, 0.31, and 0.31 μM, respectively [105,106]. A pair of epimeric polyketides, microketides A and B (**223** and **224**), were isolated from the fungus *Microsphaeropsis* sp. RA10-14; and **223** exhibited potential antibacterial activity against *P. aeruginosa*, *N. brasiliensis*, *K. rhizophila*, and *Bacillus anthraci* with the same MIC value of 0.19 μg/mL [107]. A novel metabolite was characterized from the fungus *Aspergillus fumigatus* 14–27 and identified as *epi*-pinophilin B (**225**) by spectroscopic and chemical methods [108]. From soft coral-derived fungus *Stachybotrys chartarum* SCSIO41201, four new polyketide derivatives, named arthproliferins A–D (**226–229**), were isolated; and compound **226** was found to show weak inhibitory activity against MRSA ATCC 29213 (MIC = 78 µg/mL) [81]. Polyketide pigment produced by *Talaromyces* spp. has been reported as a non-toxic red Monascus-like azaphilone pigment. Therefore, polyketides were not only a class of bioactive compounds, but also have been characterized as a source of red pigments [109] (Figure 11).

### 5.2. Anthraquinone

Cultivation of *Penicillium* sp. SCSGAF 0023 yielded a new polyketide, 6,8,5’6’-tetrahydroxy-30 methyl-flavone (**230**), and three known analogs emodin (**231**), citreorosein (**232**), and isorhodoptilometrin (**233**). Compounds **230**–**232** displayed significant antifouling activity against *Balanus amphitrite* larvae settlement with EC_50_ values of 6.7, 6.1, 17.9, and 13.7 μg/mL, respectively [110]. A new benzopyranone namely coniochaetone K (**234**), together with six known compounds, named coniochaetone A (**235**), 8-hydroxy-6-methylxanthone-1-carboxylic acid (**236**), methyl 8-hydroxy-6-methyl-9-oxo-9H-xanthene-1- carboxylate (**237**), methyl 8-hydroxy-6-(hydroxymethyl)-9-oxo-9H-xanthene-1-carboxylate (**238**), 8-(methoxycarbonyl)-1-hydroxy-9-oxo-9H-xanthene-3-carboxylic acid (**239**), and 3,8-dihydroxy-6-methyl-9-oxo-9H-xanthene-1-carboxylate (**240**), were obtained from the Beibu Gulf-derived coral symbiotic fungus *Cladosporium halotolerans* GXIMD 02502. Moreover, all compounds **234**–**240** at a concentration of 10 μM revealed outstanding cytotoxicity against two human prostatic cancer cell lines, C4-2B and 22RV1, with an inhibitory rate ranging from 55.8% to 82.1%. Among them, compound **240** was found to be the strongest with inhibitions of 82.1% and 77.7%, respectively [111]. In addition, compound **240** also showed activity against cell lines K562, HL-60, HeLa, and BGC-823 at a concentration of 100 μg/mL with an inhibition rate of 36.1%, 62.4%, 13.9%, and 11.4%, respectively [112]. Three metabolites were identified as 12-dimethoxypinselin (**241**), 12-*O*-acetyl-AGI-B4 (**242**), and AGI-B4 (**243**); from solid rice cultures of the marine-derived fungus *Scopulariopsis* sp., which was collected from the Red Sea hard coral *Stylophora* sp.. Compound **243** showed significant cytotoxicity against L5178Y mouse lymphoma cells (IC_50_ = 1.5 µM) [34]. A new anthraquinone derivative macrosporin 2-*O*-α-*D*-glucopyranoside (**244**), together with two known analogues altersolanol B (**245**) and altersolanol A (**246**), were obtained from the fungus *Stemphylium lycopersici*. Compound **245** exhibited evident activities against HCT-116 and MCF-7 cancer cell lines, with the IC_50_ values of 3.5 and 9.0 μM, respectively, while **246** showed IC_50_ values of 1.3 and 7.2 μM, respectively. Moreover, compound **246** also exhibited growth inhibition against Huh7 cancer stem cell-like cells (IC_50_ = 38.0 μM) [113]. In addition, compound **246** exhibited cytotoxic, cytostatic, anti-inflammatory, and anti-migrative activity against K562 and A549, surprisingly, while it had no effect on normal cells [114]. Wang et al. isolated 20 metabolites from coral-associated fungus. A number of new compounds were derived from the mycelia of the *Alternaria* sp. ZJ-2008003 strain isolated from a *Sarcophyton* sp. soft coral in South China Sea, including tetrahydroaltersolanols C−F (**247**–**250**) and dihydroaltersolanol A (**251**). The results of the antiviral activity indicated that compound **247** exhibited activity against PRRSV (IC_50_ = 65 μM) [115]. Investigation of the *Penicillium chrysogenum* derived from gorgonian resulted in the isolation of a new flavone, penimethavone A (**252**), and exhibited middling anticancer activity against HeLa and rhabdomyosarcoma cell lines with IC_50_ values of 8.41 and 8.18 μM, respectively [116]. *Aspergillus candidus* derived from the gorgonian coral *Anthogorgia ochracea* collected from the South China Sea produced a new compound, aspergivone B (**253**), which showed weak inhibition against α-glucosidase (IC_50_ = 244 μg/mL) [117]. Two new hydroxyanthraquinones, harzianumnones A (**254**) and B (**255**), together with seven known analogues (**256–262**), namely pachybasin (**256**), chrysophanol (**257**), frangulaemodin (**258**), phomarin (**259**), (+)-20*S*-isorhodoptilometrin (**260**), 1-hydroxy-3-hydroxymethylanthraquinone (**261**), and *Ω*-hydroxydigitoemodin (**262**) were discovered from the soft coral-associated fungus *Trichoderma harzianum* (XS-20090075). Compounds **256**, **257**, **258**, **260**, and **262** displayed moderate AChE inhibitory effects at the concentration of 100 μM. Compounds **258**, **260**, and **261** showed weak antibacterial activity against *S. aureus* with the MIC values of 6.25, 25.0, and 25.0 μM, respectively. Compounds **260** and **261** exhibited cytotoxicity against hepatoma cell line HepG2 with IC_50_ values of 2.10 and 9.39 μM, respectively. Compound **260** was still found to reveal cytotoxicity against cervical cancer cell line HeLa (IC_50_ = 8.59 μM) [118]. For use as a bio-agricultural agent with antifungal activity against phytopathogenic fungi, pachybasin (**256**) induced toxicity in zebrafish embryos in a dose-dependent manner. In addition, chrysophanol (**257**) displayed activity against human malignancy of colorectal cancer [119,120]. Three described analogues and a new anthraquinone derivative, named nidurufin (**263**), 8-*O*-methylaverufin (**264**), averufanin (**265**), and 8-*O*-methylnidurufin (**266**), were produced by fungus *Aspergillus* sp. derived from the gorgonian coral *Dichotella gemmacea.* Furthermore, nidurufin (**263**) exhibited excellent inhibition against K562 and HL-60 cell lines with IC_50_ values of 0.87 and 1.46 µM, respectively. Meanwhile, 8-*O*-methylnidurufin (**266**) and 8-*O*-methylaverufin (**264**) showed antibacterial activity against *Micrococcus luteus* with the same MIC values of 6.25 µM. Moreover, **264** was also active against *Mucor miehei* and **265** exhibited inhibitory effects on ACAT1 and ACAT2 with IC_50_ values of 28 and 12.1 µM, respectively [121,122,123] (Figure 12).

From a *Penicillium* sp. SCSGAF 0023, a new compound paecilin C (**267**) and three known compounds, secalonic acid D (**268**), secalonic acid B (**269**), and penicillixanthone A (**270**), were obtained; and compounds **268**–**270** found to show medium antibacterial activity against *M. luteus*, *Pseudomonas nigrifaciens*, *E. coli*, and *B. subtilis* with MIC values ranging from 24.4 to 390.5 µg/mL. In addition, compound **268** at a concentration of 3.125 µg/mL also displayed inhibition to the growth of *Serratia onedensis* MR-1 [110]. Furthermore, **268** displayed anti-tumor activity against the K562 cell line and cytotoxic activity on the human pancreatic carcinoma PANC-1 cells accustomed to glucose-starved conditions (IC_50_ = 0.6 µM). In addition, compounds **268** and **269** revealed inhibition against *S. aureus* biofilm formation at more than 90% at 6.25 µg/mL, while **268** facilitated the development of biofilm to some extent [124,125,126]. The following seven new metabolites were found from coral-associated fungus. A new anthraquinone derivative, alterporriol Y (**271**), was obtained from the fungus *Stemphylium lycopersici* [113]. *Alternaria* sp. ZJ-2008003 yielded five novel alterporriol-type anthranoid dimers named alterporriols N−R (**272**–**276**) [115]. Besides producing polyketides, *Aspergillus* sp. collected from a gorgonian *Dichotella gemmacea* also yielded a novel compound, aspergilone B (**277**) [104]. The data showed that compound **274** exhibited cytotoxic activity for PC-3 and HCT-116 cell lines with the IC_50_ values of 6.4 and 8.6 μM, respectively, and compound **275** also showed activity against PRRSV (IC_50_ =22 μM) (Figure 13).

### 5.3. Other Aromatic Compounds

A study on fungus *Pestalotiopsis* sp. led to the separation of four compounds, pestalone (**278**) and pestalachloride B (**279**), (±) – pestalachloride C (**280**), and (±) – pestalachloride D (**281**). Pestalone (**278**) displayed activities against MRSA, 30740, 31007, 31692, 31709, 31956, *Bacillus megaterium*, and *Micrococcus lysodeikticus* with MIC values of 12.5, 6.25, 12.5, 12.5, 12.5, 0.078, and 6.25 μM, respectively*,* while pestalachloride B (**279**) displayed moderate activities against *S. aureus* [127]. Recent biological tests indicated that **278** exhibited moderate cytotoxicity in the National Cancer Institute’s 60 human cell line and inhibited MRSA (MIC = 37 ng/mL) [128,129]. (±) - Pestalachloride D (**281**), a new chlorinated benzophenone derivative, along with a related analog (±) – pestalachloride C (**280**), was active against *E. coli*, *Vibrio anguillarum*, and *V. parahaemolyticus* with the MIC values of 5, 10, and 20 μM, respectively. But in the biological activity results of zebrafish, indicated compound **281** did not display any effect on a zebrafish embryo teratogenicity assay, while **280** shows abnormal growth effects [130]. *Aspergillus candidus* cultured from the gorgonian coral *Anthogorgia ochracea* collected from the South China Sea produced a novel metabolite, aspergivone A (**282**) [117]. A novel skeleton structure compound represents a class of perylene derivatives with a partially perylene quinone. They were named 7-*epi*-8-hydroxyaltertoxin I (**283**), stemphytriol (**284**), altertoxin I (**285**), stemphyltoxin II (**286**), and stemphyperylenol (**287**). These five perylene derivatives were found from the marine-derived fungi *Alternaria* sp. ZJ-2008017 collected from a soft coral *Sarcophyton* sp. Compounds **283**–**286** were obtained from bioactive experiments of the teratogenicity and lethality of zebrafish embryo and antifouling activity. The bioassay results indicated that **285** exhibited notable teratogenicity (LC_50_ = 4.54 μg/mL) and lethality (EC_50_ = 4.21 μg/mL) of zebrafish embryo and had potent antifouling activity against the barnacle *Balanus amphitrite* (IC_50_ = 0.27 μg/mL). It might be a potential antifouling agent [131]. In addition, altertoxin I (**285**) could cause DNA damage, while stemphyperylenol (**287**) was found to be active against *E. coli*, *S. aureus*, and multi-drug-resistant *P. aeruginosa* [132,133,134]. Naphthalenedicarboxylic acid (**288**), a novel compound, was isolated from the fungus *Xylaria* sp. FM1005, discovered from *Sinularia densa* [57]. The soft-coral-derived fungus *Alternaria alternata* L3111′A was the source of a new perylenequinone-related compound, alternatone A (**289**), and a known perylenequinone named alterperylenol (**290**). Alternatone A (**289**) displayed anticancer activity against the human hepatoma carcinoma HepG-2 cell line. In addition, alterperylenol (**290**) exhibited cytotoxicity against A-549, HCT-116, and HeLa cell lines with IC_50_ values of 2.6, 2.4, and 3.1 μM, respectively [135,136]. Four new oligophenalenone dimers were yielded from the soft coral-associated fungus *Talaromyces verruculosus* named verruculosins A–B (**291**, **292**), bacillisporin F (**293**), and xenoclauxin (**294**). Among which, compounds **291**, **293**, and **294** displayed middle inhibitory activity against CDC25B with IC_50_ values of 0.38, 0.40, and 0.26 µM, respectively. Furthermore, bacillisporin F (**293**) also exerted antibacterial activity against *S. aureus* with a MIC value of 15.6 µg/mL [137,138] (Figure 14).

Two new chromanones named phomalichenones H, J (**295**, **296**) and a described compound phomalichenone D (**297**) were obtained from the coral-associated fungus *Parengyodontium album* SCSIO 40430. Compound **297** revealed inhibition against MRSA shhs-A1, with MIC values of 32–64 μg/mL [139]. A new griseofulvin derivative, eupenigriseofulvin (**298**), was yielded from the EtOAc extract of *Eupenicillium* sp. SCSIO41208 [140]. 3,7-dihydroxy-1,9-dimethyl dibenzofuran (**299**) was obtained from a soft coral-derived fungus *Talaromyces* sp. SCSIO 041201. The biological assay indicated that compound **299** displayed antifouling activity against *Bugula neritina* larva with a LC_50_ value of 3.06 μg/mL; and it showed moderate antibacterial activities against *E. coli*, MRSA, *S. aureus*, and *E. faecalis*, with MIC values ranging from 0.45 to 15.6 μg/mL, respectively [141]. Two known analogues named phomalichenone A (**300**) and phomalichenone B (**301**) were isolated from the coral-derived fungus *Parengyodontium album* sp. SCSIO 40430, and both of them showed inhibitory activity against MRSA shhs-A1 and *Mycobacterium tuberculosis* H37Ra with MIC values of 64, 64, 16, and 64 μg/mL, respectively [139]. Six new citrinin analogues penicitols A (**302**), E−I (**303**–**307**), were isolated from a coral-associated fungus *Penicillium citrinum*, of which **302** exhibited cytotoxic activity against K562, A549, HL-60, BEL-7402, MCE-7, HCT-116, and MDA-MB-231 tumor cells, with IC_50_ values of 8.8, 28, 20, 12, 15, 24, and 26 μM, respectively. Moreover, **307** showed cytotoxic activity against A549 and BEL-7402, with IC_50_ values of 19 and 17 μM, respectively. Compounds **302** and **307** displayed antibacterial activity against *S. aureus*, *B. subtilis*, and Vancomycin-resistant *E. faecalis* 1010798 (VRE), with MIC values of 16, 32, 32 and 64, 32, 32 μM, respectively [142]. A new polyphenol, talaversatili A (**308**), was isolated from a soft coral-derived fungus *Talaromyces* sp. SCSIO 041201 [141]. The strain *Dichotomomyces* sp. L-8 associated with the soft coral *Lobophytum crissum* produced two new compounds, dichotones A (**309**) and B (**310**) [82]. Territrem A (**311**) was isolated and identified from a coral-derived fungus *Aspergillus terreus*. Compound **311** was active against NO production with significant inhibitory potency (IC_50_ =29.34 μM) [24]. Two new sulfur-containing benzofuran derivatives, eurothiocins A (**312**) and B (**313**), were produced by the fungus *Eurotium rubrum* SH-823, which was gathered from a *Sarcophyton* sp. soft coral collected in the South China Sea. Compounds **312** and **313** showed strong inhibitory effects against α-glucosidase with IC_50_ values of 17.1 and 42.6 μM, respectively. A recent study showed that **312** displayed anti-neuroinflammatory effects [143,144]. Shao et al. found eight novel metabolites and seven known compounds from coral-derived fungi, which were identified as 3′-OH-tetrahydro-auroglaucin (**314**) and (3′*S**,4′*R**)-6-(3′,5-epoxy-4′-hydroxy-1′-heptenyl)-2-hydroxy-3-(3′′-methyl-2′′-butenyl) benzaldehyde (**315**), talaromycins A–C (**316**–**318**), phomaethers A–C (**319**–**321**); and tenellic acid A methyl ester (**322**), sulochrin (**323**), (–)-*bis*-dechlorogeodin (**324**), isodihydroauroglaucin (**325**), flavoglaucin (**326**), 2,4-diphenyldichloroasterric acid (**327**), and geodin (**328**), respectively. Two novel metabolites, 3′-OH-tetrahydro-auroglaucin (**314**) and (3′*S**,4′*R**)-6-(3′,5-epoxy-4′-hydroxy-1′-heptenyl)-2-hydroxy-3-(3′′-methyl-2′′-butenyl) benzaldehyde (**315**) were gathered from a fungus *Eurotium* sp. [145]. Three novel diphenyl ether derivatives, talaromycins A–C (**316**–**318**), together with a known metabolite tenellic acid A methyl ester (**322**), were obtained from fungus *Talaromyces* sp. In addition, compound **318** exerted inhibition against the larval settlement of the barnacle *Balanus amphitrite* with an EC_50_ value of 2.8 μg/mL. Furthermore, compound **322** showed outstanding anticancer activity against HepG2, Hep3B, MCF-7/ADR, PC-3, and HCT-116 with IC_50_ values of 4.3, 9, 8.2, and 9.8 μM, respectively. [146]. Another three new compounds, phomaethers A–C (**319**–**321**) were obtained from a gorgonian-derived fungus *Phoma* sp. (TA07-1), while compounds **319** and **321** exhibited evident antibacterial activity against five pathogenic bacteria (*S. albus, S. aureus, E. coli, V. parahaemolyticus, V. anguillarum*) with MIC values ranging from 0.312 to 10 μM [147]. Examination of the fungus *Aspergillus* sp. associated with coral led to the separation of six known compounds. Both sulochrin (**323**) and (–)-bis-dechlorogeodin (**324**) isolated from *Aspergillus* sp. derived from soft coral were active against *V. anguillarum*, *Aeromonas salmonicida*, and *P. aeruginosa* with MICs of 15.06, 15.15, 7.53 μM and 30.12, 30.3, and 3.78 μM, respectively. Moreover, **324** also revealed moderate activity against Jurkat, A549, and HeLa cells with IC_50_ values of 10.69, 10.69, and 3.56 μM, respectively [148]. Two known benzaldehyde compounds, isodihydroauroglaucin (**325**) and flavoglaucin (**326**), exhibited evident antiviral activity against HSV-1 with EC_50_ values of 4.73 and 6.95 μM, respectively. In addition, flavoglaucin (**326**) displayed considerable activity against DPPH (IC_50_ = 11.3 μM) [149,150]. Two known derivatives were isolated from soft coral *Sinularia* sp. derived fungus *Aspergillu*s sp., namely 2,4-diphenyldichloroasterric acid (**327**) and geodin (**328**). In the bioassays, compound **327** displayed inhibition to *S. aureus* (MIC = 12.5 μM) and **328** exhibited cytotoxic activity against six human cancer cell lines, including BT474, NCI-H460, H-1975, K562, DU145, and A549, with IC_50_ values of 8.88, 9.22, 9.96, 11.14, 14.44, and 11.05 μM, respectively. Meanwhile, geodin (**328**) was also considered as a potential semisynthesized compound for the purpose of producing derivatives with better insecticidal activities [151,152]. The fungus *Scopulariopsis* sp. obtained from the Red Sea hard coral *Stylophora* sp. resulted in the discovery of two phenyl ethers violaceols I and II (**329** and **330**). Moreover, compounds **329** and **330** showed remarkable cytotoxicity toward the growth of L5178Y with IC_50_ values of 9.5 and 9.2 µM, respectively, while the positive control was 4.5 µM (kahalalide F). In addition, both of them displayed antimicrobial activities against *Staphylococcus saprophyticus*, *S. aureus*, MRSA, *B. subtilis*, *B. cereus, Salmonella typhimurium, Shigella sonnei, and C. albicans* (MIC < 9.765–312.5 μg/mL); and they were also found to act as actin inhibitors inducing cell shape elongation in fibroblast cells [34,153,154]. The chemical investigation of *Parengyodontium album* SCSIO 40430 led to three new chromanones, phomalichenones K-M (**331–333**) and two known metabolites, 8-ethyl-5,7-dihydroxy-2-methylchroman-4-one (**334**), and (±)-trieusol D (**335**). Compounds **334** and **335** showed inhibition against MRSA shhs-A1, with the same MIC value of 64 μg/mL, and they were also active against *M. tuberculosis* H37Ra with MIC values of 32 and 64 μg/mL, respectively [139] (Figure 15).

## 6. Lactones

A known ten-membered macrolide named (3*Z*,5*S*,6*E*,8*S*,9*S*,10*R*)-8-chloro-5,8,9,10-tetrahydro-5,9-dihydroxy-10-methyl-2H-oxecin-2-one (**336**) was obtained from a soft coral-derived fungus *Lophiostoma* sp. ZJ-2008011 [36]. The bioassay test showed that **336** exhibited antibacterial activities against *B. cereus* (MIC = 3.12 μM). *Aspergillus versicolor* LCJ-5-4 yielded a new lactone, marked as versicolactone A (**337**) [85]. A recent study indicated that sorbicillinoid derivatives from marine-derived fungus also possessed radical scavenging activities. In the last four years, 69 new sorbicillinoids were identified from fungi [155,156]. Dimethyl incisterol A3 (**338**) and (l7*R*)-17-methylincisterol (**339**) were obtained from a soft coral-derived fungus *Talaromyces* sp. SCSIO 041201. Both **338** and **339** showed antifouling activities to *Bugula neritina* larva, with LC_50_ values of 3.13 and 4.15 μg/mL, respectively [141]. Four new butanolide derivatives 8*’’R*,9*’’-*diolversicolactone B (**340**), 8*’’S*,9*’’-*diolversicolactone (**341**), 3’-isoamylene butyrolactone IV (**342**), and 4’-dehydroxy aspernolide A (**343**), together with two described compounds named butyrolactone I (**344**) and versicolactone B (**345**), were produced by a coral-associated fungus *Aspergillus terreus*. They were tested for the inhibition of NO production in RAW264.7 mouse macrophages induced by LPS at a concentration of 20 μM (positive control group indomethacin of 50 μM). The inhibitory effect of compound **345** was significantly stronger than that of indomethacin. In addition, compounds **342** and **344** also displayed a modest inhibitory effect on NO production with inhibition ratios of nearly 25.1% and 25.3%, respectively [157]. A new compound named versicolactone G (**346**) was isolated from a coral-associated fungus *Aspergillus terreus*, which showed remarkable anti-inflammatory activity against NO production (IC_50_ = 15.72 μM). Other studies indicated that butyrolactone I (**344**) presented with antitumor activity against HL-60 with an IC_50_ value of 13.22 μM and ameliorated AlCl_3_-induced cognitive deficits in zebrafish in a dose-dependent manner [24,158,159]. (5*S*,6*S*)-dihydroxylasiodiplodin (**347**) was produced by the fungus *Pseudallescheria ellipsoidea* F42-3 derived from the soft coral *Lobophytum crissum* [51]. Four described metabolites, 2-*O*-Methylbutyrolactone I (**348**), 2-*O*-Methylbutyrolactone II (**349**), demethoxycarbonylbutyrolactone II (**350**), and butyrolactone III (**351**), were obtained from fungus *Aspergillus* sp. Compounds **349**–**351** showed inhibition against the settlement of barnacle *Balanus amphitrite* with EC_50_ values of 2.10, 4.25, and 2.89 µg/mL, respectively. Furthermore, compounds **348** and **349** as well exhibited outstanding antibacterial activities toward *S. aureus, S. epidermidis,* and *B. cereus, V. parahaemolyticus,* and *V. anguillarum* with MIC values ranging from 1.56 to 12.5 µM [160]. Three novel metabolites, cochliomycins A-C (**352**–**354**), along with six known analogues, diacetyl derivative (**355**), monoacetyl derivative (**356**), zeaenol (**357**), LL-Z1640-1 (**358**), LL-Z1640-2 (**359**), and paecilomycin F (**360**), were isolated from the fungus *Cochliobolus lunatus* from the gorgonian *Dichotella gemmacea*. In addition, compounds **353**, **355**–**358**, and **360** exhibited antifouling activity against the larval settlement of the barnacle *Balanus amphitrite* with EC_50_ values of 1.2, 15.4, 12.5, 5.0, 5.3, and 17.9 µg/mL, respectively. Moreover, compound **359** showed inhibition to HgCl_2_-induced JNK phosphorylation at 5–100 ng/mL [161,162]. The chemical exploration of *Aspergillus* sp. SCSGAF 0076 led to the isolation of three novel compounds, aspergillides A-C (**361**–**363**). Furthermore, compound **363** exerted considerable antifouling activity against *Bugula neritina* larvae settlement with LC_50_/EC_50_ > 25 [101,163,164]. A known metabolite satratoxin F (**364**) was discovered from the soft coral-derived fungus *Stachybotrys chartarum* SCSIO41201, and **364** showed weak inhibitory activity against MRSA ATCC 29213 with the MIC value of 39 µg/mL. Furthermore, compound **364** displayed excellent cytotoxic activities against five human cancer cell lines (MDA-MB-231, C4-2B, MGC803, MDA-MB-468, and A549) with IC_50_ values less than 39 nM [81]. Five new mycophenolic acid derivatives, 6-(5-carboxy-3-methylpent-2-enyl)-7-hydroxy-3,5-dimethoxy-4-methylphthalan-1-one (**365**), 6-(5-methoxycarbonyl-3-methylpent-2-enyl)-3,7-dihydroxy-5-methoxy-4-methylphthalan-1-one (**366**), 6-(3-carboxybutyl)-7-hydroxy-5-methoxy-4-methylphthalan-1-one (**367**), 6-(5-(2,3-dihydroxy-1-carboxyglyceride)-3-methylpent-2-enyl)-7-hydroxy-5-methoxy-4-methylphthalan-1-one (**368**), and 6-(5-(1-carboxy-4-*N*-carboxylate)-3-methylpent-2-enyl)-7-hydroxy-5-methoxy-4-methylphthalan-1-one (**369**); and five already reported metabolites 8-*O*-methyl mycophenolic acid (**370**), 3-hydroxymycophenolic acid (**371**), *N*-mycophenoyl-*L*-valine (**372**), *N*-mycophenoyl-*L*-phenyloalanine (**373**), and *N*-mycophenoyl-*L*-alanine (**374**) were obtained from the coral-derived fungus *Penicillium bialowiezense*. Moreover, compounds **365**–**374** were tested for immune inhibitory activity; and compounds **372**, **372** and **366** showed significant inhibition to IMPDH2 with IC_50_ values ranging from 0.84 to 0.95 µM; while **366–369** and **372–374** revealed notable inhibitory potency with IC_50_ values ranging from 3.27 to 24.68 µM [165]. A new azaphthalide derivative, (*S*)-3-hydroxy-2,7-dimethylfuro [3,4-b] pyridin-5(7H)-one (**375**), and a new phthalide derivative, (*S*)-7-hydroxy-3-((*S*)-1-hydroxyethyl) isobenzofuran-1-(3H)-one (**376**), together with two known compounds (*R*)-3-hydroxymellein (**377**) [166] and (3*R*,4*S*)-trans-4-hydroxymellein (**378**) [167], were obtained from the coral-associated fungus *Aspergillus* sp. SCSIO41405. The isolated compounds were tested for antibacterial and enzyme inhibition activities. Compound **377** revealed weak antibacterial activity against methicillin-resistant *Staphylococcus epidermidis*, and **378** also exhibited antibacterial activity against *E. faecalis*, with the same MIC value of 100 μg/mL [168]. Ten novel enantiomers (±)-eurotiumides A-E (**379**–**383**) were isolated from a gorgonian-derived fungus *Eurotium* sp. XS-200900E6. In the bioassays, (+)- and (-)-eurotiumides B (**380**) and D (**381**) displayed antifouling activities against the larval settlement of the barnacle *Balanus amphitrite* with the EC_50_ values ranging from 0.7 to 2.3 μg/mL, and compounds **379–382** also showed potent antibacterial activities [169]. A novel scopupyrone (**384**) was produced by the solid rice cultures of the marine-derived fungus *Scopulariopsis* sp. collected from the Red Sea hard coral *Stylophora* sp. [34] (Figure 16).

## 7. Steroids

Four new compounds, arthriniumsteroids A−D (**385**–**388**), together with a known compound named eoaspergillic acid (**389**), were obtained from the soft coral-derived fungus *Simplicillium lanosoniveum* SCSIO41212. All compounds displayed weak inhibitory activities against LPS-induced NO production in RAW 264.7 cells with inhibitory rates ranging from 21.4%−44.6% [170]. Four new steroid derivatives, penicildiones A−D (**390**–**393**), together with a described compound, stachybotrylactone B (**394**), were given by soft coral-derived fungus *Penicillium* sp. SCSIO41201. Moreover, compound **394** showed evident cytotoxic activity against HL-60, K562, MOLT-4, ACHN, 786-O, and OS-RC-2 cell lines with IC_50_ values of 5.23, 4.12, 4.31, 23.55, 7.65, and 10.81 μM, respectively [171]. Further investigation of coral-derived fungus *Aspergillus* sp. led to separation of three known compounds, melilotigenin C (**395**) [168], 14α-hydroxyergosta-4,7,22-triene-3,6-dione (**396**) [24], chaxine C (**397**), and a novel compound, 24,28-didehydro-2 (**398**) [160]. Melilotigenin C (**395**) revealed moderate inhibition to pancreatic lipase (IC_50_ = 15.6 μg/mL) [168]. **396** exhibited outstanding anti-inflammatory activity against NO production (IC_50_ = 26.83 μM) [24]. In addition, both **397** and **398** showed activities against the larval settlement of barnacle *Balanus amphitrite* with EC_50_ values of 2.50 and 18.40 μg/mL, respectively [160]. 5α,8α-*epi*dioxy-ergosta-6,22*E*-dien-3*β*-ol (**399**) was yielded by the gorgonian-derived fungus *Xylaria* sp. C-2 and displayed bioactivity against *E. coli, P. putida,* and *K. rhizophila* with MIC values of 3.13, 1.56, and 6.25 μM, respectively [172]. A novel pregnane, 3α-hydroxy-7-ene-6,20-dione (**400**) was found from a fungus *Cladosporium* sp. WZ-2008-0042 and showed significant antiviral activity against RSV (IC_50_ = 0.12 μM) [173] (Figure 17).

## 8. Other Compounds

Research on the soft coral-derived fungus *Trichoderma harzianum* (XS-20090075) led to the isolation of two new natural products, methyl-trichoharzin (**401**) and trichoharzin B (**402**), and two known compounds, eujavanicol A (**403**) and nafuredin (**404**). Compounds **401**, **403**, and **404** displayed antifouling activity against larval settlement of *Bugula neritina* with EC_50_ values of 29.8, 35.6, and 21.4 μg/mL respectively. Another study showed that eujavanicol A (**403**) revealed bioactivity against *E. coli* with a MIC value of 5.0 μg/mL [79,174]. A polysaccharide named AW1 (**405**) was isolated from the *Aspergillus ochraceus* collected from coral *Dichotella gemmacea*. AW1(**405**) was relatively rarely found in the marine metabolites. AW1 (**405**) is an extracellular polysaccharide that possesses a novel galactomannan and the molecular weight was 29 kDa [175]. Three new compounds (**406**–**408**), together with four known analogues named pseurotin A (**409**), AM6898B (**410**), and (−)-ovalicin derivative (**411**), chlovalicin (**412**), were obtained from the fungus *Pseudallescheria boydii* collected from the South China Sea soft coral *Sinularia sandensis*. Compound **409** increased the number of osteoclasts at 0.1 μM, while compound **407** decreased the numbers of osteoclasts at 1 μM; and compounds **410** and **411** increased the number of osteoclasts at both 0.1 and 1 μM. In contrast, compound **412** significantly decreased the number and reduced the area of the osteoclasts at both 0.1 and 1 μM. Pseurotin A (**409**) was active against the phytopathogenic bacteria *Erwinia carotovora* and *Pseudomonas syringae* with IC_50_ values of 220 and 112 μg/mL, respectively. It was still active against *B. cereus* and *Shigella shiga* with an MIC of 64 μg/mL. Meanwhile, it also inhibited the production of IgE with an IC_50_ value of 3.6 μM [176,177,178,179]. *Penicillium pinophilum* XS-20090E18 was the source of a new metabolite, along with three previously described compounds, hydroxypenicillide (**413**), penicillide (**414**), isopenicillide (**415**), and purpactin A (**416**). Meanwhile, all of them displayed antifouling activity against the settlement of the barnacle *Balanus amphitrite* with EC_50_ values of 6.0, 2.6, 20, and 10 μg/mL, respectively. Penicillide (**414**) was also regarded as a calcium-activated papain-like protease with an IC_50_ value of 7.1 μM. Purpactin A (**416**) was considered as an inhibitor of TMEM16A chloride channels and mucin secretion in airway epithelial cells with an approximate IC_50_ value of 2 μM; and displayed moderate inhibition against MCF7, H460, and SF268 with IC_50_ values of 20.5, 17.6, and 21.9 μM, respectively [103,180,181,182]. A novel hexahydrobenzopyran derivative, cytosporin L (**417**) and a bioactive compound, cytosporin D (**418**), were isolated from a fungus *Eutypella* sp.; and both of them exerted considerable inhibition against RSV with the IC_50_ values of 72.01 and 30.25 μM, respectively. Furthermore, compound **417** revealed bioactivity to *Micrococcus lysodeikticus* and *Enterobacter aerogenes* with the same MIC value of 3.12 μM. Cytosporin D (**418**) presented antimicrobial activity against *E. coli* and *S. aureus*. [183,184]. Two new metabolites together with a known toxin, aluminiumneoaspergillin (**419**), zirconiumneoaspergillin (**420**) and ferrineoaspergillin (**421**), were isolated from *Aspergillus* sp. SCSGAF0093. All of them exhibited lethality against brine shrimp (*Artemia salina*) with LC_50_ values of 6.61, 10.76, and 29.62 μM, respectively [77]. A new fatty acid, (2*E*,4*E*,6*S*)-6-hydroxydeca-2,4-dienoic acid (**422**), was produced by the gorgonian-associated fungus *Xylaria* sp. C-2 [172]. Cladosporilactam A (**423**), a new bicyclic lactam, was isolated from a gorgonian-derived *Cladosporium* sp. fungus and exhibited potent anticancer activity against HeLa with an IC_50_ value of 0.76 µM [185] (Figure 18).

## 9. Comprehensive Overview and Conclusions

This review concluded that coral-associated fungi are a productive source of structurally diversified secondary metabolites with various bioactivities. In particular, the present review includes 423 metabolites of newly discovered compounds and bioactive compounds with a wide range of biological activities of anticancer, antimicrobial, anti-inflammatory, anti-fouling, and other bioactivities. In addition to the current in vitro bioassays, further clinical studies of these bioactive compounds are required to determine their potential therapeutic applications.

Generally, marine natural products isolated from coral-associated fungi are dominated by aromatics (35%) and alkaloids (24%), followed by lactones, peptides, terpenes, steroids, and other compounds (Figure 19, Appendix A).

These natural compounds were isolated from several coral-derived fungi covered soft coral, gorgonian coral, hard coral, leather coral, stony coral, and some unknown ones. Among them, soft coral-derived fungi are the dominant producers of natural products, comprising more than 53% of total molecules. And the second, gorgonian-derived fungi, accounts for 37% (Figure 20, Appendix A). Generally, the epizoic relationship between symbiotic or epiphytic fungi and their host animals or environment induces the production of metabolites.

Bioactive compounds are comprised of anticancer (22%), antimicrobial (28%), anti-fouling (17%), anti-inflammatory (7%), and other biological activities (26%) (Figure 21, Appendix A). Among them, other activities include neuroprotective activity, AChE inhibitory activity, α-glucosidase activity etc. It follows that compound produced by coral-associated fungi reveal great potential bioactivities.

Structurally, aromatics (36%) and alkaloids (27%) included the major proportions of bioactive compounds, followed by lactones (11%) (Figure 22, Appendix A). What is noteworthy is that satratoxin F (**364**) as a known lactone displayed excellent cytotoxic activities against five human cancer cell lines (MDA-MB-231, C4-2B, MGC803, MDA-MB-468, and A549) with EC_50_ values less than 39 nM. A previous review indicated that fungi was regarded as the best candidate as a source of anticancer agents [186]. In addition, another alkaloid, aflaquinolone D (**111**), exhibited notable antifouling activity with EC_50_ values of 2.8 nM. Obviously, compounds containing heteroatoms exhibited more potential bioactivities.

Noteworthy is that several compounds exhibit different bioactivities. As seen in Figure 21, the biological activities mainly focus on anticancer and antimicrobial activity. Furthermore, new compounds accounted for 23%, whereas known compounds made up 77% (Figure 23). New technologies and methods should be applied to improve the discovery of new compounds and it is necessary to explore the bioactivities of known metabolites through effective biological screening methods.

In the last few decades, secondary metabolites gathered from coral-associated fungi have shown noteworthy levels in a number of clinical targets, and many of them are structurally unique and possess remarkable biological and pharmacological properties, such as anticancer and antimicrobial activity. Some of them are lead compounds and potential clinical drug candidates, such as satratoxin F with antitumor activity and aurasperone B with radical-scavenging activity against 2,2-diphenyl-1-picryl-hydrazyl. Meanwhile, marine natural products exhibit diversified bioactivity due to their special chemical structures and potential interactions with proteins.

Drug discovery from coral-associated fungi has been a sustainable and intelligent methodology that can surmount supply issues through the large-scale fermentation of fungi. Furthermore, some key biosynthetic gene clusters in the regulation of fungi expressing unique skeleton metabolites with high bioactivity remain silent [187]. Therefore, it is necessary to apply novel methods and technologies with the purpose of activating the expression of unique secondary metabolites. On the basis of simulating natural conditions, it is appropriate to add some stimulus.

In conclusion, the present review elucidated chemical structures of 423 compounds obtained from coral-associated fungi, many of them with bioactivities as promising drug lead compounds. These metabolites exhibit diversified structures and various bioactivities. In particular, some compounds revealed remarkable activity, even stronger than the positive controls. These findings indicate that these compounds have great potential in the treatment of diseases. But the bioactivity of these metabolites was mainly tested in vitro; thus, more attention should be paid to the molecular mechanism and further in vivo and preclinical studies. In conclusion, there are many coral-associated secondary metabolites with notable biological activity, and numerous drug lead compounds and new metabolites are still waiting to be discovered.

## Figures and Tables

**Figure 1 jof-08-01043-f001:**
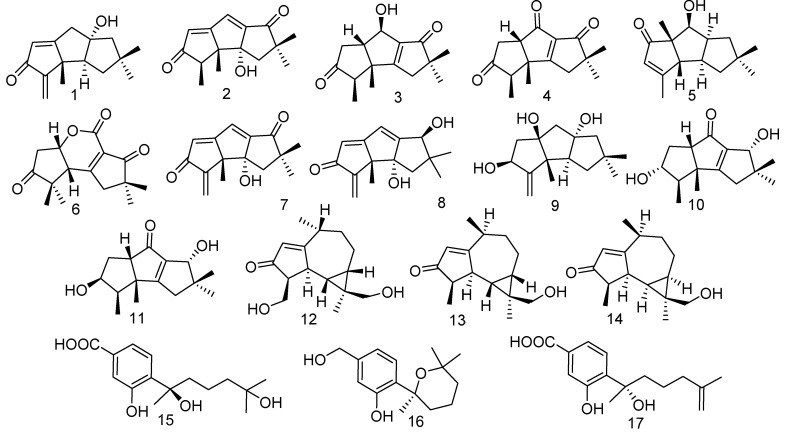
The structures of compounds **1**–**17**.

**Figure 2 jof-08-01043-f002:**
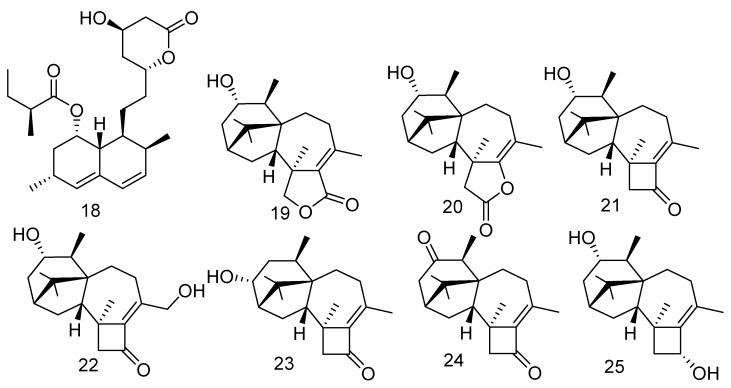
The structures of compounds **18**–**25**.

**Figure 3 jof-08-01043-f003:**
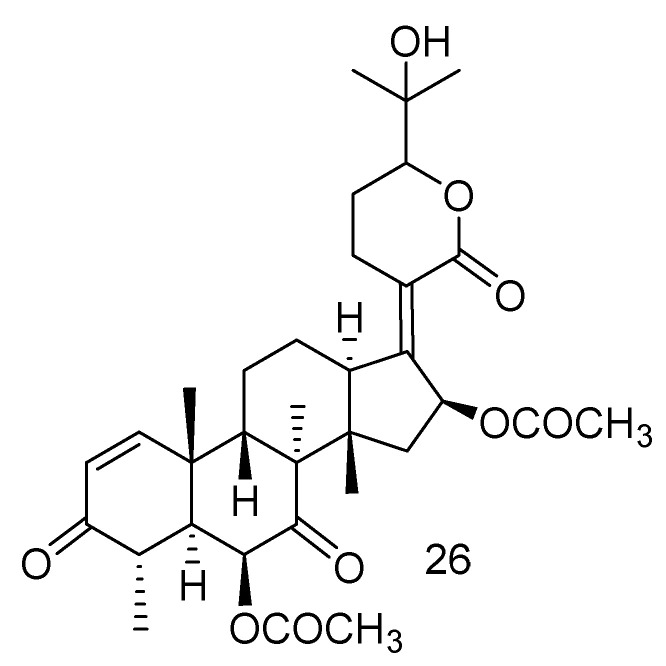
The structure of compound **26**.

**Figure 4 jof-08-01043-f004:**
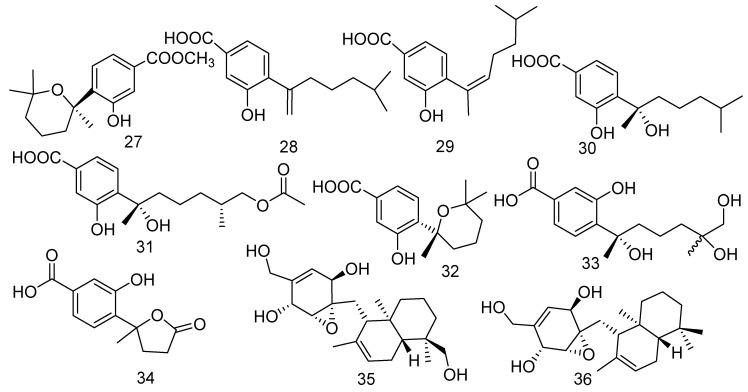
The structures of compounds **27**–**36**.

**Figure 5 jof-08-01043-f005:**
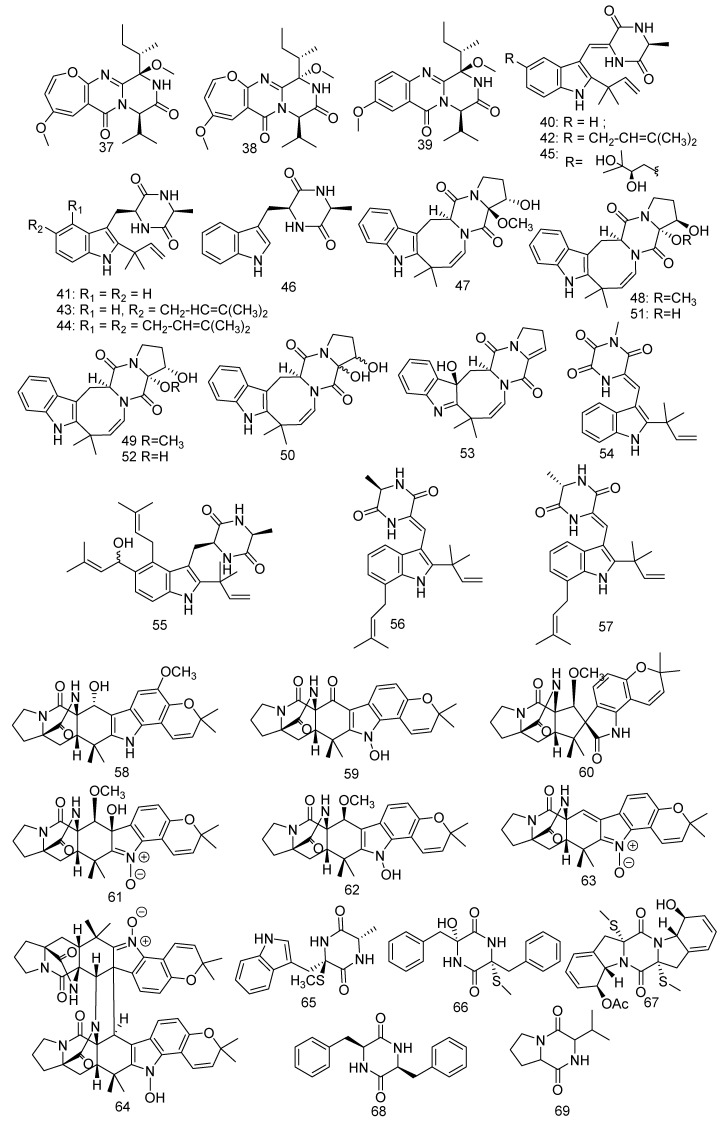
The structures of compounds **37**–**69**.

**Figure 6 jof-08-01043-f006:**
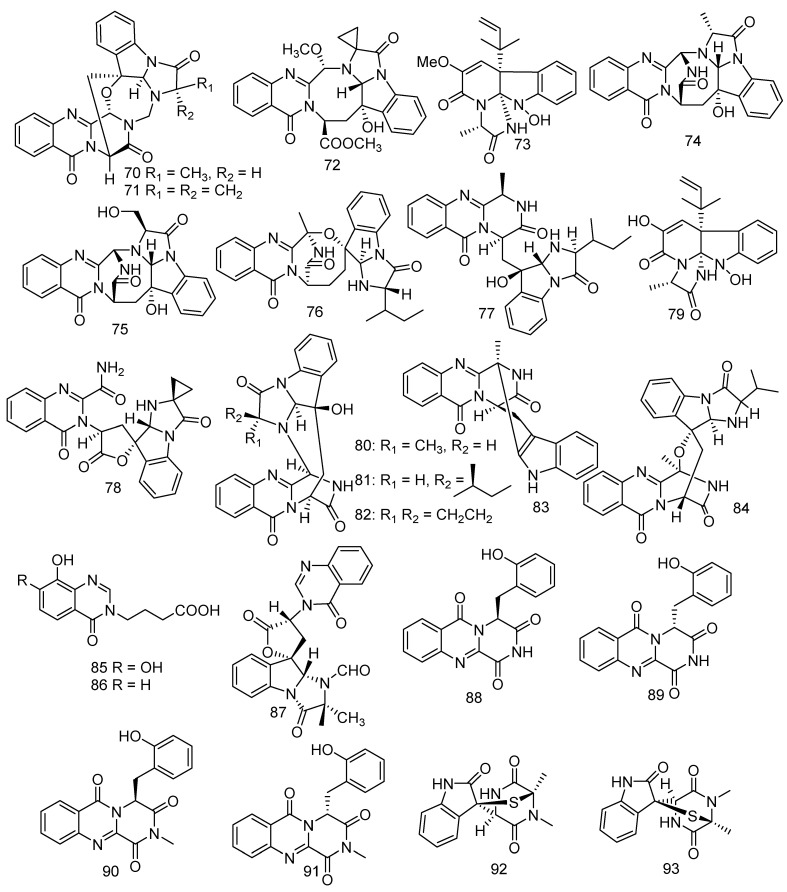
The structures of compounds **70**–**93**.

**Figure 7 jof-08-01043-f007:**
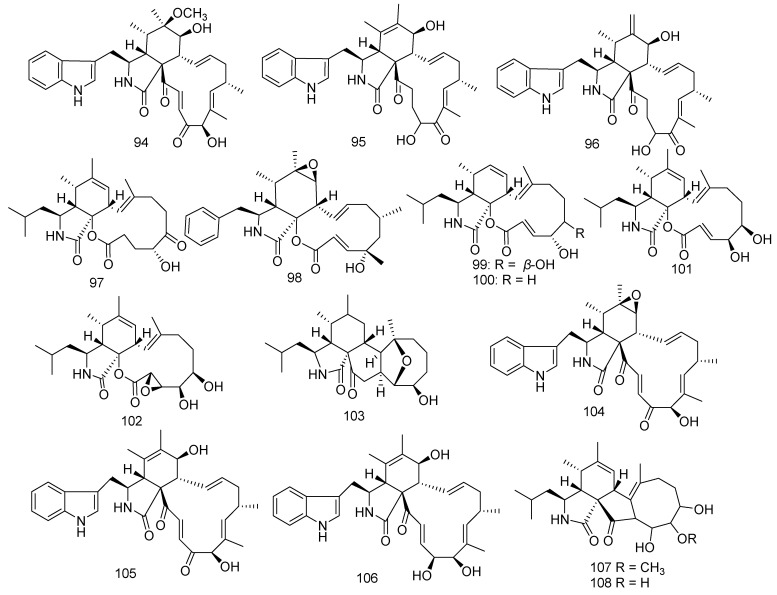
The structures of compounds **94**–**108**.

**Figure 8 jof-08-01043-f008:**
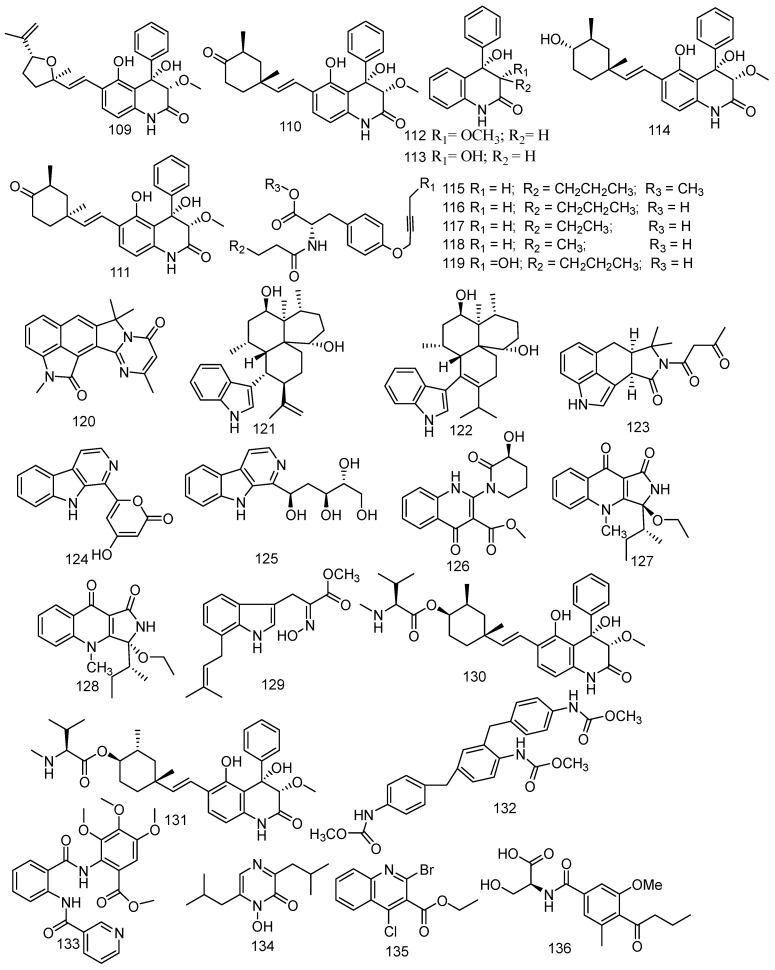
The structures of compounds **109**–**139**.

**Figure 9 jof-08-01043-f009:**
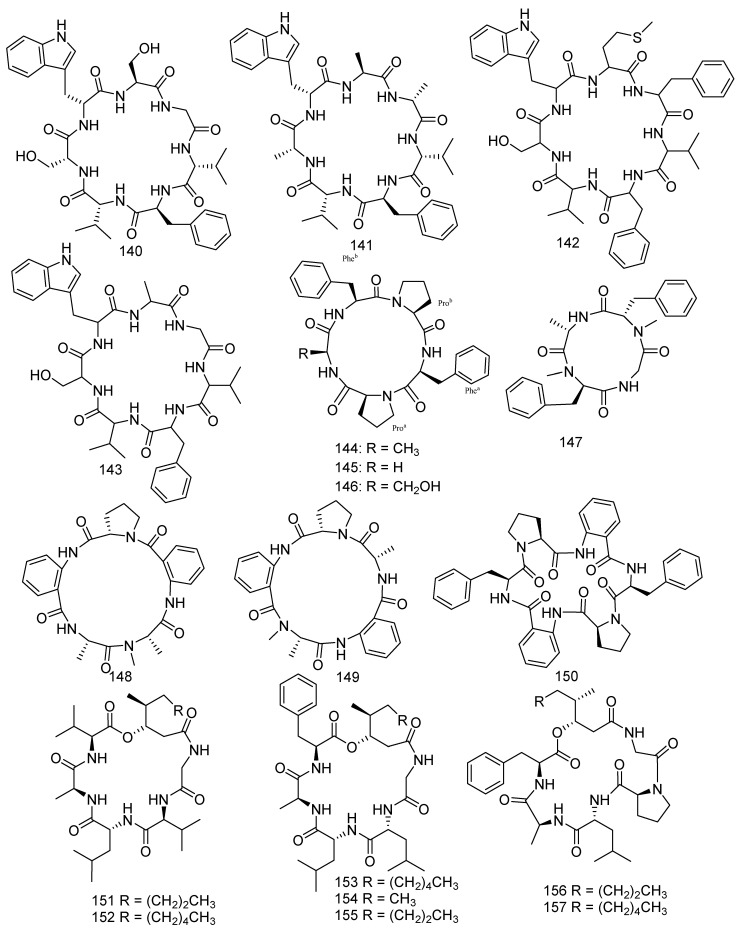
The structures of compounds **140**–**164**.

**Figure 10 jof-08-01043-f010:**
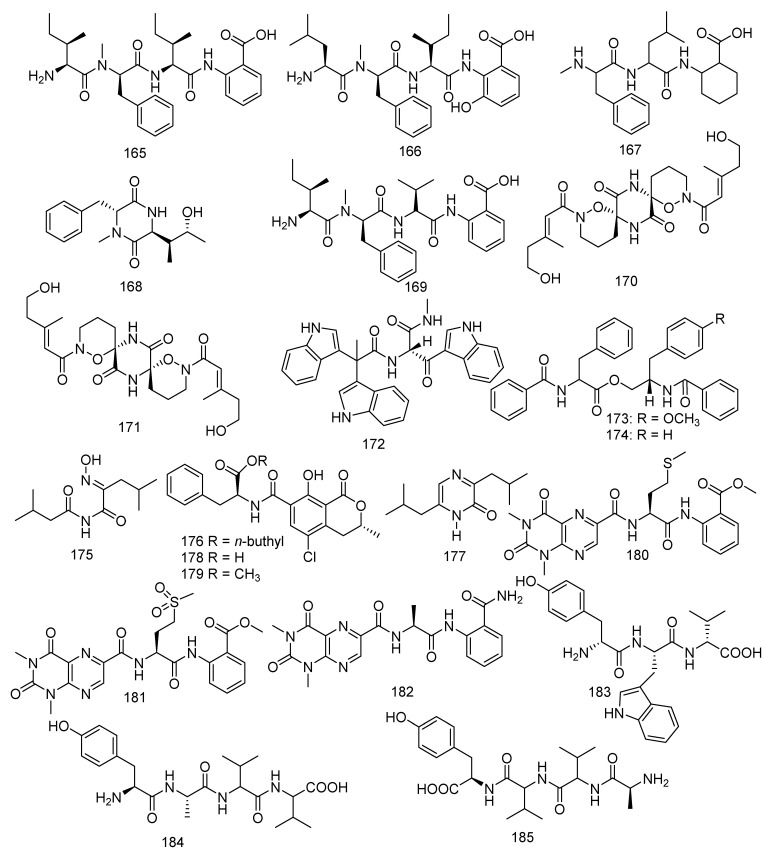
The structures of compounds **165**–**185**.

**Figure 11 jof-08-01043-f011:**
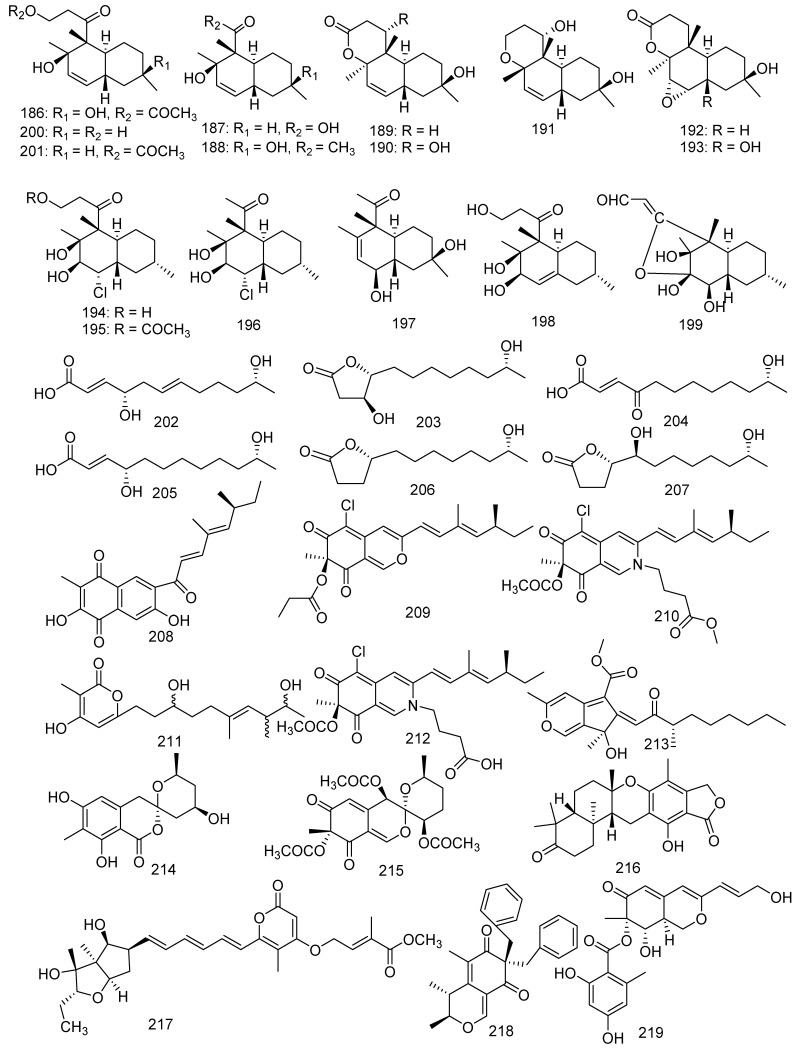
The structures of compounds **186**–**229**.

**Figure 12 jof-08-01043-f012:**
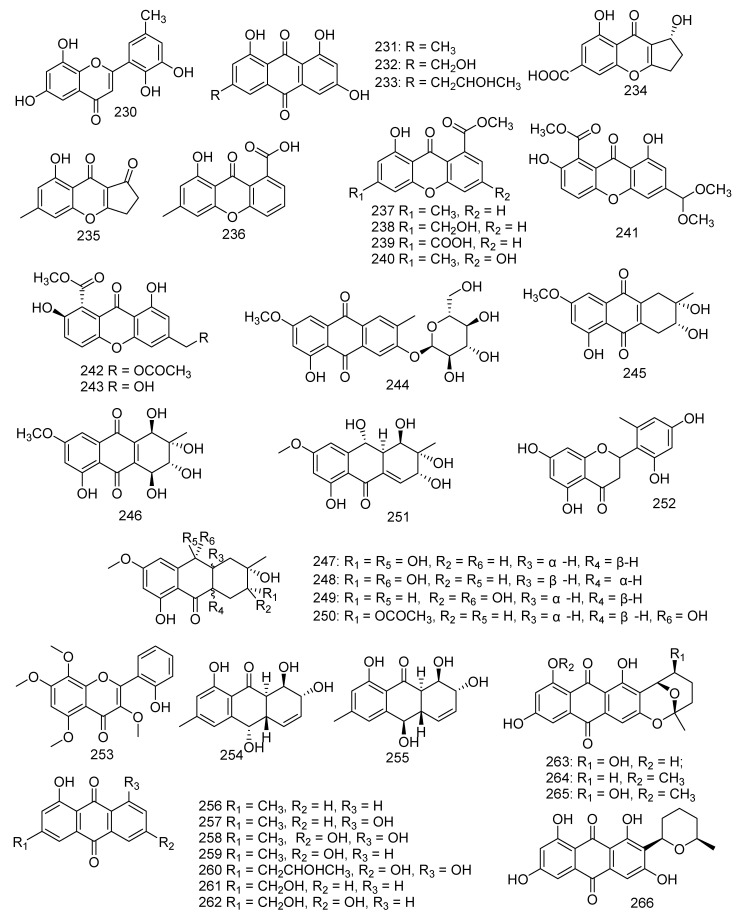
The structures of compounds **230**–**266.**

**Figure 13 jof-08-01043-f013:**
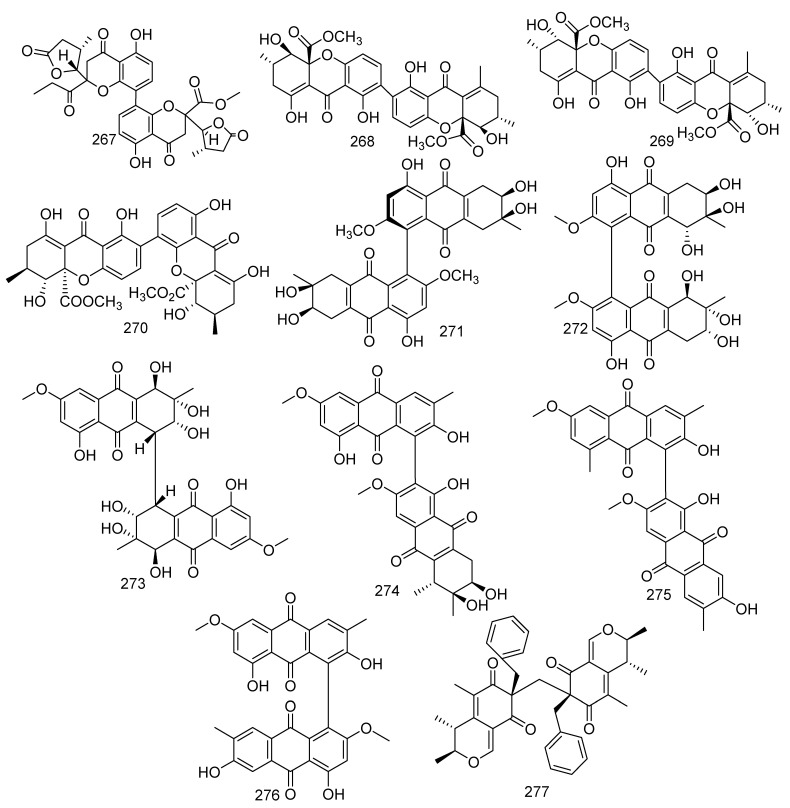
The structures of compounds **267**–**277**.

**Figure 14 jof-08-01043-f014:**
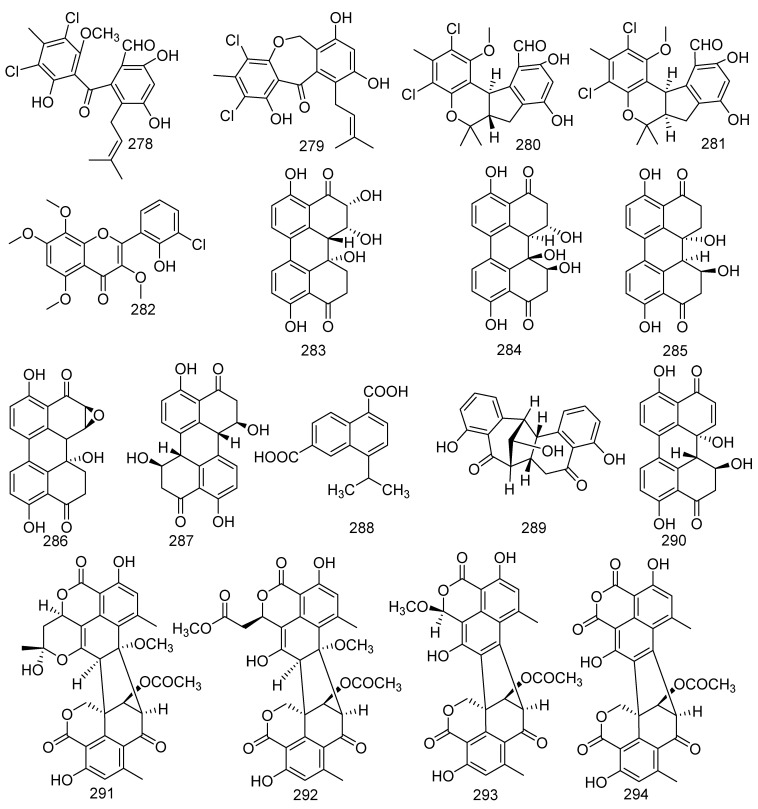
The structures of compounds **278**–**294**.

**Figure 15 jof-08-01043-f015:**
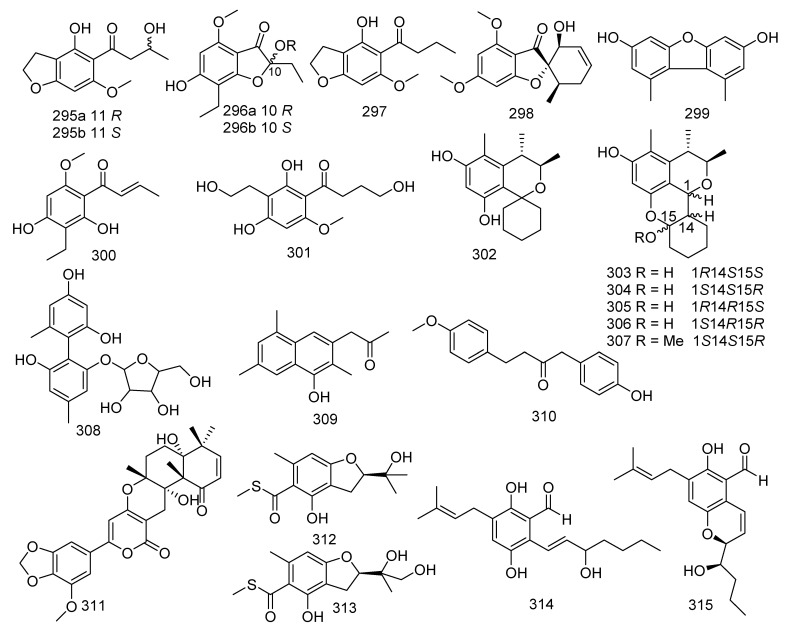
The structures of compounds **295**–**335**.

**Figure 16 jof-08-01043-f016:**
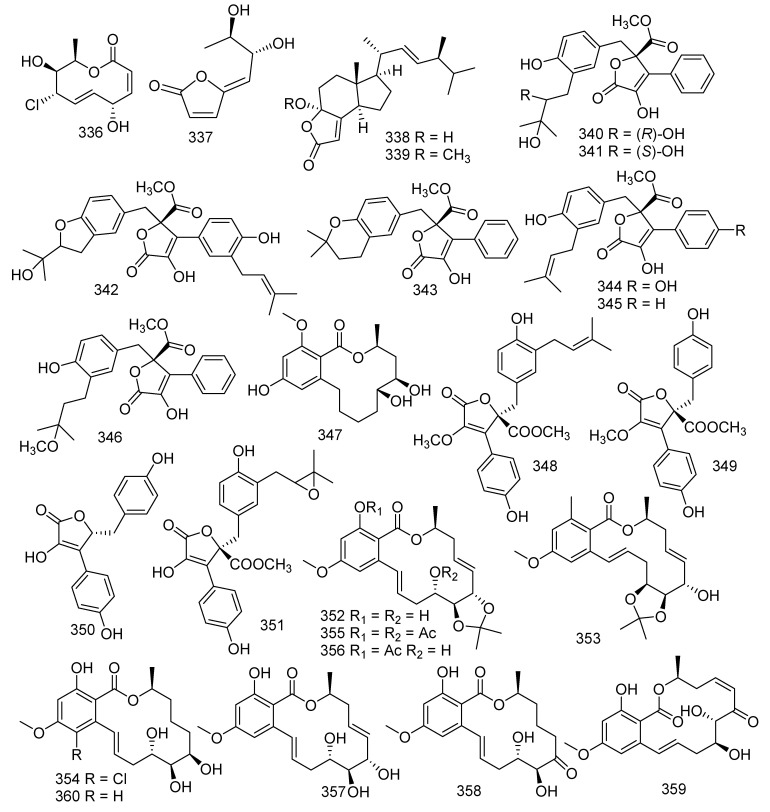
The structures of compounds **336**–**384**.

**Figure 17 jof-08-01043-f017:**
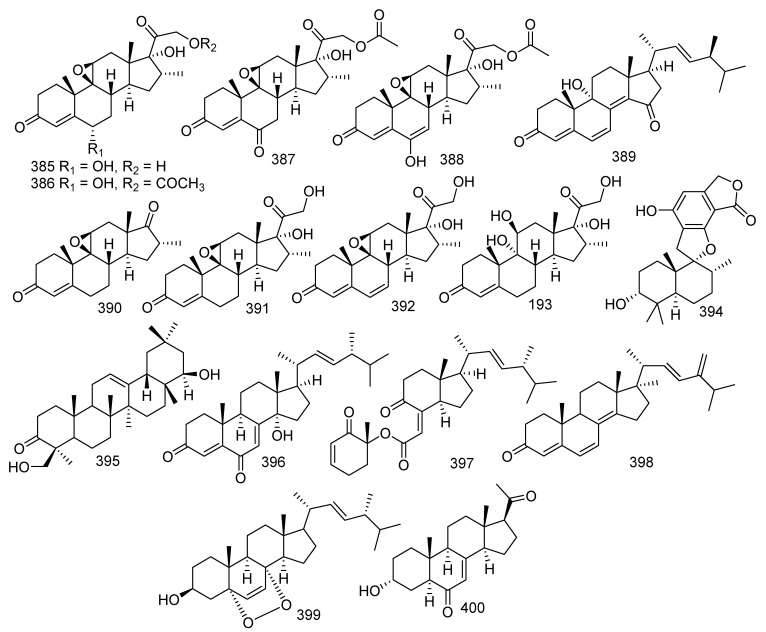
The structures of compounds **385**–**400**.

**Figure 18 jof-08-01043-f018:**
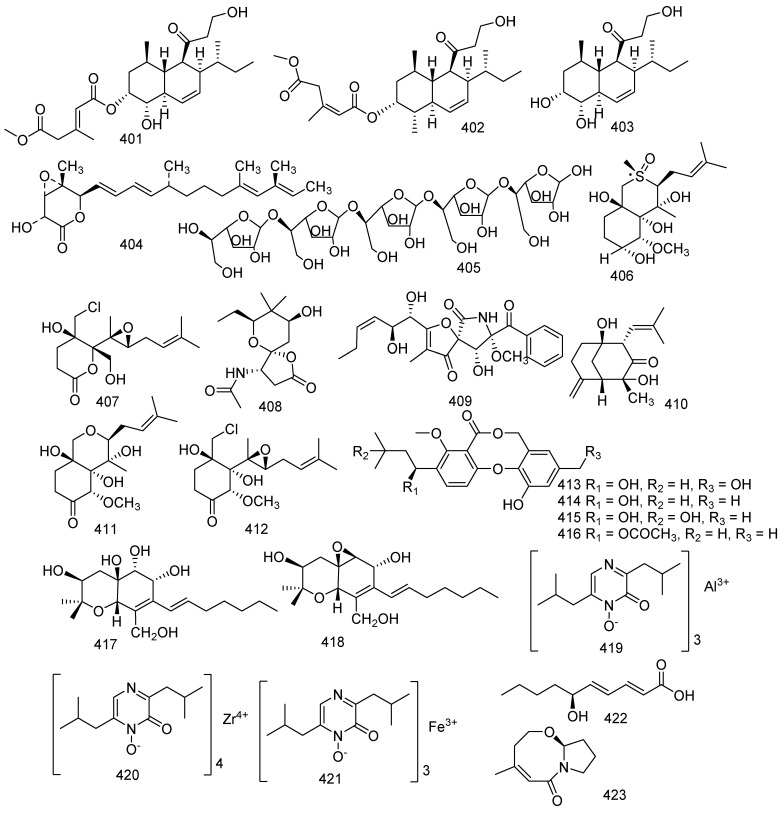
The structures of compounds **401**–**423**.

**Figure 19 jof-08-01043-f019:**
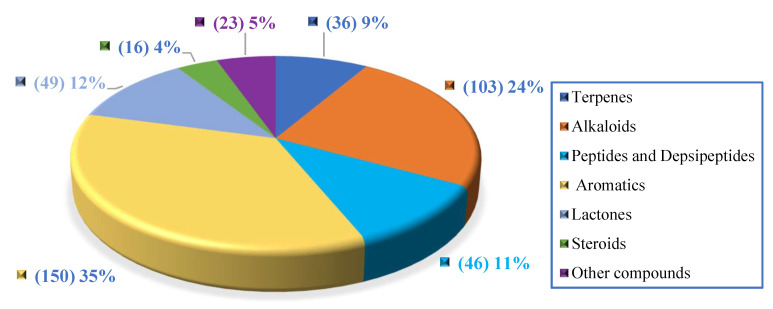
Distribution of the compounds according to chemical structure.

**Figure 20 jof-08-01043-f020:**
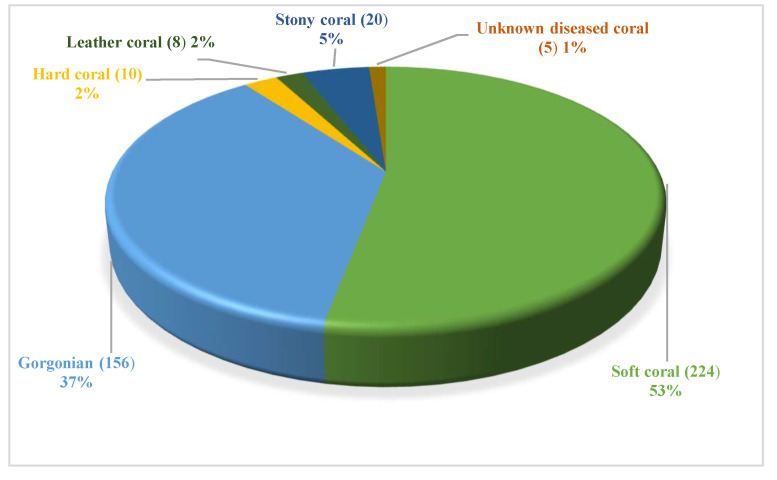
The strain source of the natural products from coral-derived fungi.

**Figure 21 jof-08-01043-f021:**
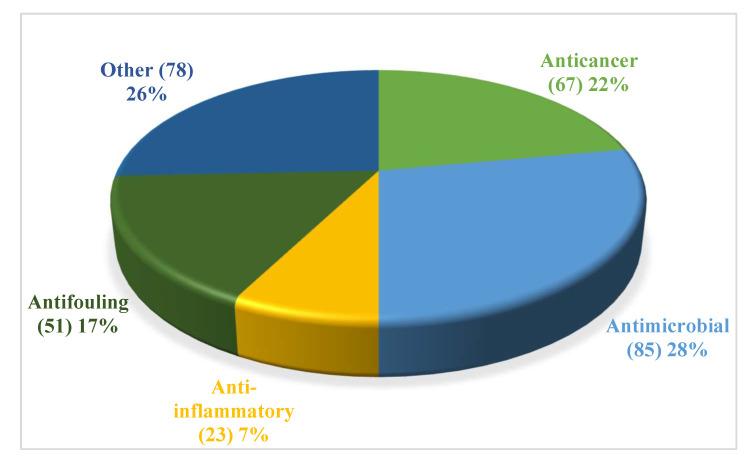
Distribution of the compounds according to bioactivities.

**Figure 22 jof-08-01043-f022:**
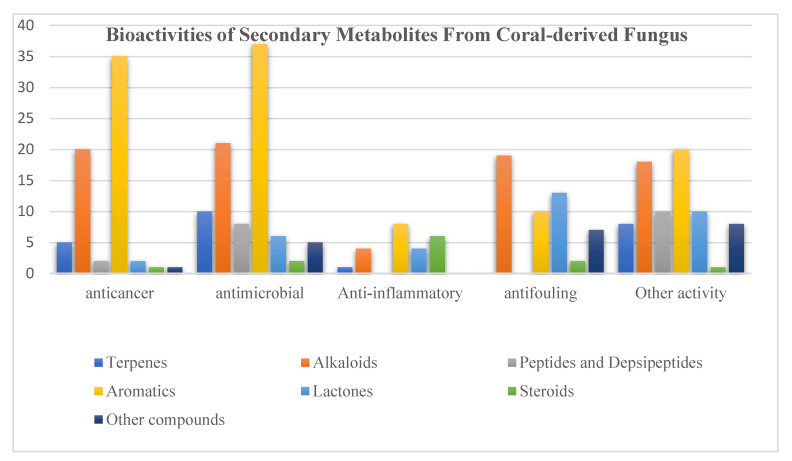
Distribution of the bioactive compounds according to structures.

**Figure 23 jof-08-01043-f023:**
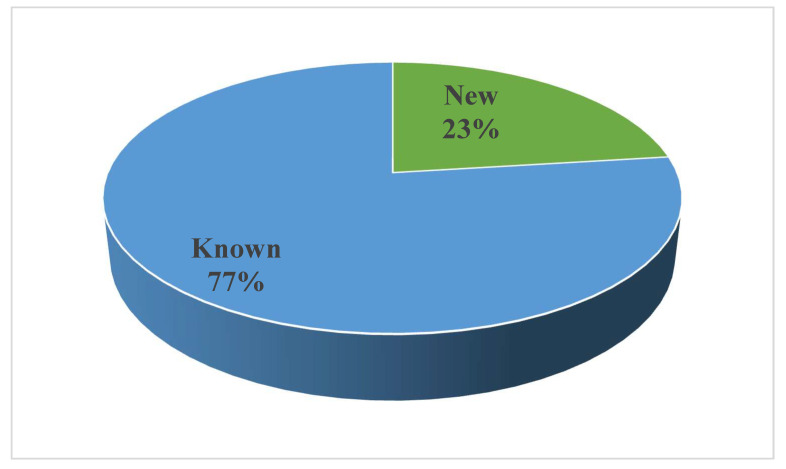
Distribution of the bioactive compounds between new compounds and known compounds.

## Data Availability

Not applicable.

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
