# Peer review of "Secondary Metabolites from Coral-Associated Fungi: Source, Chemistry and Bioactivities"

_jof, 2022, doi:10.3390/jof8101043_

Round 1

Reviewer 1 Report

The submitted work is a review comprising all compounds isolated from coral-associated fungi and their biological activity from 2010 to 2021. The text is well written and is a great contribution to the growing field of microbial marine natural products. 

However, a few compounds isolated from the actinomycetes genus Streptomyces were included. Streptomyces are gram-positive bacteria and not fungi. Therefore the compounds isolated from this genus are not the scope of the review and should be removed. Compounds from Streptomyces are described in the following parts:

-lines 323-324: compounds 136-137

-lines 598-600: compounds 307-308

-lines 874-877: compounds 455-456

After these minor revisions, I recommend the publication of the review in the Journal of Fungi.

Author Response

However, a few compounds isolated from the actinomycetes genus Streptomyces were included. Streptomyces are gram-positive bacteria and not fungi. Therefore the compounds isolated from this genus are not the scope of the review and should be removed. Compounds from Streptomyces are described in the following parts:

-lines 323-324: compounds 136-137

-lines 598-600: compounds 307-308

-lines 874-877: compounds 455-456

After these minor revisions, I recommend the publication of the review in the Journal of Fungi.

Response: Thank you very much for your letter. We are grateful for the suggestion. Based on your comment, we have removed these compounds (136-137, 307-308, 455-456) isolated from the actinomycetes genus Streptomyces.

Reviewer 2 Report

In the manuscript, the authors have mentioned that the secondary metabolites of coral-associated fungi lead to a valuable but extra-large chemical database. Considering the quality of assays and the overall results, the paper is suitable for publication in the Journal of Fungi with some revisions, as reported below.

1.      In vitro should be in italic (lines 105, 256, 660)

2.      40 & 42 (line 175 and others). Could you check if spelling 1&2 is accepted by the journal's rules?

3.      IC50 values must be standardized, IC50 =XX.XµM or XX.X ±X.XX µM.

4.      The structures need to be revised, especially the bond angles. Structures: 27, 32, 73, 120-127, 133, 134

5.      Authors must follow a standard for scientific names for organisms (biological assays). If it is the first citation in the text, the scientific name must have the genus and species. From the first citation, the genus must be abbreviated, followed by the species.

a.      Line 99: Staphylococcus. aureus, Bacillus cereus, Kocuria rhizophila, Pseudomonas putida, Peseudomonas aeruginosa, Salmonella enterica, Nocardia brasiliensis

b.      Line 136: G. graminis. Gaeumannomyces graminis (?)

c.      Line 141: S. albus Staphyloccocus albus

d.      Line 153: Bacillus cereus, B. cereus

e.      Line 237: Bacillus subtilis, S.taphylococcus albus, and Vibrio parahemolyticus

f.       Line 268: Balanus Amphitrite or Balanus amphitrite?

g.      Line 268: S.taphylococcus albus, S. aureus, E.scherichia coli and B.acillus cereus

h.       Line 301: staphylococcus aureus, S. aureus

i.       Line 326: S.taphylococcus aureus

j.       Line 334: E. coli, A. Baumannii, P. aeruginosa, K. pneumonia, MRSA, and E. faecalis, C. albicans, should be correct.

k.      Line 342: S. aureus, S. epidermidis (complete name, first time in text), B. subtilis, B. dysenteriae, B. proteus, E. coli

l.       Line 351: K. rhizophila (complete name, first time in text) and S. aureus

m.     Lines: 468,472 and 478, 518, 560, 587, 588, 592, 594, 612, 615,650, 723-729, 740, 760, 769, 773, 776, 779, 802, 826, 828, 840, 853,858, 869, 878, please standardize scientific names

6.      Line 175: “….DPPH scavenging effect with IC50 44.30±0.06 and 103 μM,…”… DPPH scavenging effect with IC50 44.30±0.06 and 103 μM, respectively….

7.      Line 182: TMEM16A with 65.0% inhibition (5 mu g/ml) [46]. Is it 5 mu g/ml, correct?

8.      Line 258: HL60, Hela, and MOLT-4, with IC50 values of 13.53±1.90, 8.9, 5.9, 1.4, 9.2, 2.1, 1.4, 8.2, 2.5, 2.8, and 1.4 μM, respectively. Please standardize IC values

9.      Line 261: “….exhibited inflammatory activity….” Inflammatory activity, is it correct?

10.   Line 442: 11-hydroxy-c-dodecalactone (224), is it correct?

11.   Line 528: IC50 values of 8.41 and 8.18 µM [120]. Respectively?

12.   Line 530: alpha-glucosidase or α-glucosidade?

13.   Line 575: ZJ-2008003 yielded Five. ZJ-2008003 yielded five

14.   Line 620 : “…Aalterperylenol (317) also…”

15.   Line 831: What is the virus family?

16.   Line 873: Brine shrimp, would it be Artemia salina? Suggestion: “…all of them exhibited lethality against brine shrimp (Artemia salina) with LC50 values of 6.61…”

17.   Can the captions of figures 19, 20, 21, and 23 be improved?

Author Response

Response to Reviewers 2

  1. In vitroshould be in italic (lines 105, 256, 660)

Response: Thank you for your suggestion, we amended the relevant part in this manuscript.

  1. 40 & 42 (line 175 and others). Could you check if spelling 1&2 is accepted by the journal's rules?

Response: We apologize for the language problems in the original manuscript. We have revised and modified according to the information suggested by the reviewer.

  1. IC50values must be standardized, IC50 =XX.XµM or XX.X ±X.XX µM.

Response: Our deepest gratitude goes to you for your careful work and suggestions that have helped improve this paper substantially. We have standardized the IC50 values in the manuscript.

  1. The structures need to be revised, especially the bond angles. Structures: 27, 32, 73, 120-127, 133, 134

Response: Thank you for your precious comments and advice. According to references, we have been revised the structures of 27, 32, 73, 120-127; while structures of 133-134 were in accordance of its reference.

  1. Authors must follow a standard for scientific names for organisms (biological assays). If it is the first citation in the text, the scientific name must have the genus and species. From the first citation, the genus must be abbreviated, followed by the species.

Response: Thanks for your suggestion We have standardized scientific names of organisms in biological assays.

  1. Line 175: “….DPPH scavenging effect with IC50 44.30±0.06 and 103 μM,…”… DPPH scavenging effect with IC50 44.30±0.06 and 103 μM, respectively….

Response: Thank you very much. We have revised the manuscript accordingly

  1. Line 182: TMEM16A with 65.0% inhibition (5 mu g/ml) [46]. Is it 5 mu g/ml, correct?

Response: We thank the reviewer’s effort reading our manuscript and providing us with useful comments. On the basis of carefully comparing of relative reference ,5 mu g/ml is correct.

  1. Line 258: HL60, Hela, and MOLT-4, with IC50 values of 13.53±1.90, 8.9, 5.9, 1.4, 9.2, 2.1, 1.4, 8.2, 2.5, 2.8, and 1.4 μM, respectively. Please standardize IC values

Response: Thanks for your suggestion. We have standardized the IC50 values in the manuscript.

  1. Line 261: “….exhibited inflammatory activity….” Inflammatory activity, is it correct?

Response: We are grateful for the suggestion. We apologize for the careless omission. We have replaced “inflammatory” with “anti-infllammatory”

  1. Line 442: 11-hydroxy-c-dodecalactone (224), is it correct?

Response: We agree with the comment and revised the compound’name 11-hydroxy-γ-dodecalactone according to the relative reference.

  1. Line 528: IC50 values of 8.41 and 8.18 µM [120]. Respectively?

Response: We are grateful for the suggestion. Yes, the IC50 values of 8.41 and 8.18, respectively.

  1. Line 530: alpha-glucosidase or α-glucosidade?

Response: Thank you very much. We replaced alpha-glucosidase to α-glucosidase.

  1. Line 575: ZJ-2008003 yielded Five. ZJ-2008003 yielded five

Response: Thank you for the suggestion. According to the reviewer’s comment, we have revised the manuscript.

  1. Line 620 : “…Aalterperylenol (317) also…”

Response: Thanks very much. We have revised the sentence.

  1. Line 831: What is the virus family?

Response: Thank you for your precious comments and advice. We have added the detailed virus family.

  1. Line 873: Brine shrimp, would it be Artemia salina? Suggestion: “…all of them exhibited lethality against brine shrimp (Artemia salina) with LC50 values of 6.61…”

Response: We deeply appreciate the reviewer’s suggestion. According to the reviewer’s comment, we have replaced the “Brine shrimp” to “brine shrimp (Artemia salina)”.

  1. Can the captions of figures 19, 20, 21, and 23 be improved?

Response: We deeply appreciate the reviewer’s suggestion. We carefully improved the captions of 19, 20, 21, and 23.

Reviewer 3 Report

The manuscript by Ying Chen and co-workers " Secondary Metabolites from Coral-associated Fungi: Source, Chemistry and Bioactivities” explores extensively the recent literature (2010- 2021) in relation to the description of structures and biological activities of new metabolites from marine-derived fungi associated to Coral.

The manuscript explores adequately the impressive amount of published works but it presents several criticisms. For instance, some of the claimed metabolites in the paper are not derived from fungi, and a number of the phrases are ambiguous or only provide partial information. The English has to be accurately revised.

In addition, the manuscript presents various mistakes, both substantial and clerical and needs to undergo an extensive review.

The paper is unsuitable for publication on Journal of Fungi in its present form.

GENERAL OBSERVATIONS

-          An accurate control of taxonomic names is needed. For each taxon the complete scientific name could be reported:  the specific epithet must be preceded by a generic name, written out in full the first time (i.e. see line 99)

-          The unit measure must be correct and uniform in the text i.e.

o   mL and L or ml and l

o   µM/L is NOT correct to indicate a concentration (see line 197)

-          The following word must be reported in the same manner in the text:

o   Co-worker or coworker

o   Anti-viral or antiviral

-          Please check all acronyms and uniform in the text  (i. e, LoVo line 89 / lovo line 91….)

-          Mention the significance of acronyms in the first citation.

-          In the manuscript several metabolites derived from bacteria were reported; these could be removed from the review:

o   Compounds 120-131 l are derived from Vibrio (reference 74)

o   Compounds 152-154 are not from coral derived fungi they should be removed from the text

o   Compounds 366-367 derived from actinomycete and deep seawater

-          Line 653: “A new compound paecilospirone (335) was derived from marine-derived fungi Paecilomyces sp. collected from tropical and sub-tropical coral reef environments”….

-          The sentences is ambiguous Two strains of Paecilomyces sp. were examined? Both produced 335 compound?

In the reference work is clearly indicated that compound 335 is derived from Paecilomyces sp. Isolated from the Coral reef at Yap.

For additional suggestions please see the enclosed PDF file.

Author Response

Dear editor/review:

I am writing this letter to resubmit my paper, entitled “Secondary Metabolites from Coral-associated Fungi: Source, Chemistry and Bioactivities”. According to the reviewer’s comments, we revised the manuscript, the revised parts were highlighted in red.

If you have any question or suggestion with this manuscript, please contact me. Thank you very much for your attention and consideration.

Best Wishes,

Bin Yang

   An accurate control of taxonomic names is needed. For each taxon the complete scientific name could be reported:  the specific epithet must be preceded by a generic name, written out in full the first time (i.e. see line 99)

Response: Thanks for your suggestion. We have revised these taxonomic names.

-          The unit measure must be correct and uniform in the text i.e.

o   mL and L or ml and l

o   µM/L is NOT correct to indicate a concentration (see line 197)

-          The following word must be reported in the same manner in the text:

o   Co-worker or coworker

o   Anti-viral or antiviral

-          Please check all acronyms(缩写) and uniform in the text  (i. e, LoVo line 89 / lovo line 91….)

-          Mention the significance of acronyms in the first citation.

-          In the manuscript several metabolites derived from bacteria were reported; these could be removed from the review:

o   Compounds 120-131 l are derived from Vibrio (reference 74)

o   Compounds 152-154 are not from coral derived fungi they should be removed from the text

o   Compounds 366-367 derived from actinomycete and deep seawater

 Response: We have unified “Co-worker” to “ coworker”, “mL and L”to “ml and l” “ Anti-viral” to “antiviral” and already removed these compounds inconsistent with this paper.

-          Line 653: “A new compound paecilospirone (335) was derived from marine-derived fungi Paecilomyces sp. collected from tropical and sub-tropical coral reef environments”….

-          The sentences is ambiguous Two strains of Paecilomyces sp. were examined? Both produced 335 compound?

Response: Thank you very much. We have compared to the reference and removed this compound because it was inconsistent with this paper.

In the reference work is clearly indicated that compound 335 is derived from Paecilomyces sp. Isolated from the Coral reef at Yap.

For additional suggestions please see the enclosed PDF file.

Response:  We deeply apologized for the mistakes in this manuscript. We have revised this paper point by point according to your PDF file.

Round 2

Reviewer 3 Report

Dear Authors,

 A more accurate revision is needed before the publications of the manuscript.

Some of the previously indicated and new mistakes are still present in the text and an extensive editing of the English language is required.

- Some species names are not in italic, some specific epithets are reported with capitalized first letter (see attached file) 

-  The measure of concentration reported as microM/l is not correct . The Molarity is a concentration measure

- In several sentences mu M is reporeted,  this is not a scientific annotation. 

Author Response

Dear editor/review:

I am writing this letter to resubmit my paper, entitled “Secondary Metabolites from Coral-associated Fungi: Source, Chemistry and Bioactivities”. According to the reviewer’s comments, we revised the manuscript, the revised parts were highlighted in red.

If you have any question or suggestion with this manuscript, please contact me. Thank you very much for your attention and consideration.

Best Wishes,

Bin Yang

South China Sea Institute of Oceanology, Chinese Academy of Sciences,

No. 164, West Xingang Road,

Guangzhou 510301, China

Tel: 86-20-89023174 / E-mail: yangbin@scsio.ac.cn

 A more accurate revision is needed before the publications of the manuscript.

Some of the previously indicated and new mistakes are still present in the text and an extensive editing of the English language is required.

- Some species names are not in italic, some specific epithets are reported with capitalized first letter (see attached file) 

Response:  Thanks for your suggestion. We have revised these species name.

-  The measure of concentration reported as microM/l is not correct . The Molarity is a concentration measure

Response: Thank you very much. We have corrected the concentration unit according to relative references.

- In several sentences mu M is reporeted, this is not a scientific annotation. 

Response: We have revised the “mu M” to “μM” through carefully comparing the literature data.
